# A geometric shape regularity effect in the human brain

**Mathias Sablé-Meyer[1,2,3]\*, Lucas Benjamin[2], Cassandra Potier Watkins[1], Chenxi He[2], Maxence Pajot[2], Théo Morfoisse[2], Fosca Al Roumi[2], Stanislas Dehaene[1,2]**

[1]Collège de France, Université Paris-Sciences-Lettres (PSL), Paris, France; [2]Cognitive Neuroimaging Unit, CEA, INSERM, Université Paris-Saclay, NeuroSpin Center, Paris, France; [3]Sainsbury Wellcome Centre for Neural Circuits and Behaviour, University College London, London, United Kingdom

## eLife Assessment

This **important** series of studies provides converging results from complementary neuroimaging and behavioral experiments to identify human brain regions involved in representing regular geometric shapes and their core features. Geometric shape concepts are present across diverse human cultures and possibly involved in human capabilities such as numerical cognition and mathematical reasoning. Identifying the brain networks involved in geometric shape representation is of broad interest to researchers studying human visual perception, reasoning, and cognition. The evidence supporting the presence of representation of geometric shape regularity in dorsal parietal and prefrontal cortex is **solid**, but does not directly demonstrate that these circuits overlap with those involved in mathematical reasoning. Furthermore, the links to defining features of geometric objects and with mathematical and symbolic reasoning would benefit from stronger evidence from more fine-tuned experimental tasks varying the stimuli and experience.

**\*For correspondence:**
mathias.sable-meyer@ucl.ac.uk

**Competing interest:** The authors declare that no competing interests exist.

## Abstract

The perception and production of regular geometric shapes, a characteristic trait of human cultures since prehistory, has unknown neural mechanisms. Behavioral studies suggest that humans are attuned to discrete regularities such as symmetries and parallelism and rely on their combinations to encode regular geometric shapes in a compressed form. To identify the brain systems underlying this ability, as well as their dynamics, we collected functional MRI in both adults and 6-year-olds, and magnetoencephalography data in adults, during the perception of simple shapes such as hexagons, triangles, and quadrilaterals. The results revealed that geometric shapes, relative to other visual categories, induce a hypoactivation of ventral visual areas and an overactivation of the intraparietal and inferior temporal regions also involved in mathematical processing, whose activation is modulated by geometric regularity. While convolutional neural networks captured the early visual activity evoked by geometric shapes, they failed to account for subsequent dorsal parietal and prefrontal signals, which could only be captured by discrete geometric features or by bigger deep-learning models of vision. We propose that the perception of abstract geometric regularities engages an additional symbolic mode of visual perception.

## Introduction

Long before the invention of writing, the very first detectable graphic productions of prehistoric humans were highly regular non-pictorial geometric signs such as parallel lines, zig-zags, triangular, or checkered patterns (*Henshilwood et al., 2018*; *Waerden, 2012*). Human cultures throughout

the world compose complex figures using simple geometrical regularities such as parallelism and symmetry in their drawings, decorative arts, tools, buildings, graphics, and maps (*Tversky, 2011*). Cognitive anthropological studies suggest that, even in the absence of formal western education, humans possess intuitions of foundational geometric concepts such as points and lines and how they combine to form regular shapes (*Dehaene et al., 2006*; *Izard et al., 2011*). The scarce data available to date suggests that other primates, including chimpanzees, may not share the same ability to perceive and produce regular geometric shapes (*Close and Call, 2015*; *Dehaene et al., 2022*; *Sablé-Meyer et al., 2021*; *Saito et al., 2014*; *Tanaka, 2007*), though unintentional-but-regular mark-marking behavior has been reported in macaques (*Sueur, 2025*). Thus, studying the brain mechanisms that support the perception of geometric regularities may shed light on the origins of human compositionality and, ultimately, the mental language of mathematics. Here, we provide a first approach through the recording of functional MRI and magneto-encephalography signals evoked by simple geometric shapes such as triangles or squares. Our goal is to probe whether, over and above the pathways for processing the shapes of images such as faces, places, or objects, the regularities of geometric shapes evoke additional activity.

The present brain-imaging research capitalizes on a series of studies of how humans perceive quadrilaterals (*Sablé-Meyer et al., 2021*). In that study, we created 11 tightly matched stimuli that were

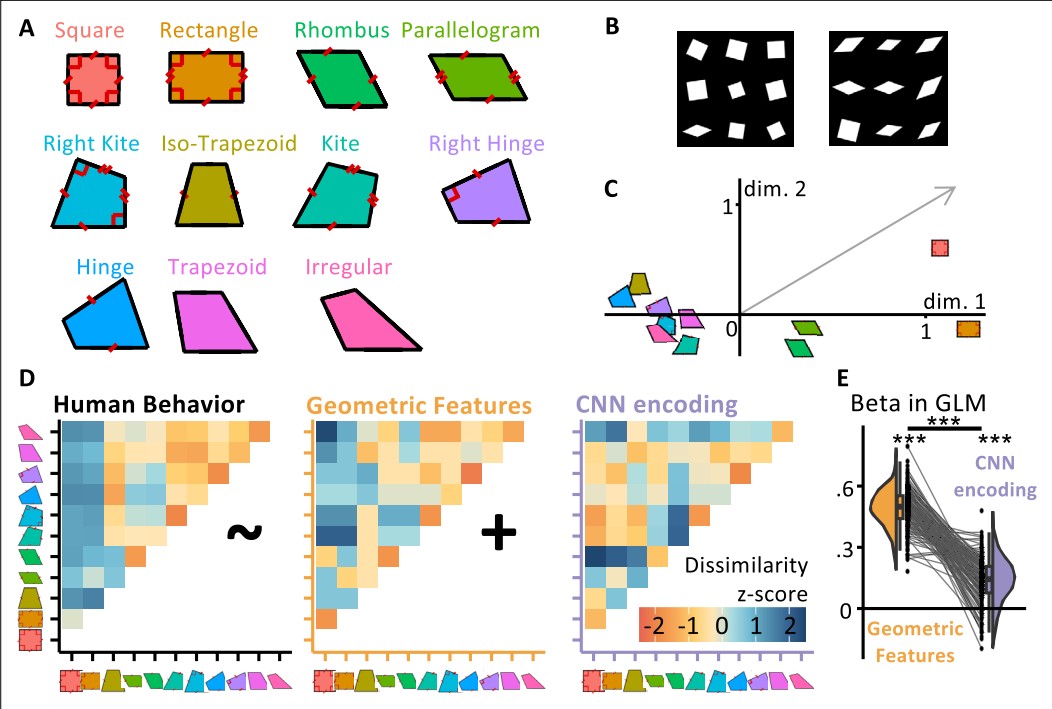

**Figure 1.** Measuring and modeling the perceptual similarity of geometric shapes. (**A**) The 11 quadrilaterals used throughout the experiments (colors are consistently used in all other figures). (**B**) Sample displays for the behavioral visual search task used to estimate the 11 × 11 shape similarity matrix. Participants had to locate the deviant shape. The right insert shows two trials from the behavioral visual search task, used to estimate the 11 × 11 shape similarity matrix. Participants had to find the intruder within nine shapes. (**C**) Multidimensional scaling of human dissimilarity judgments; the gray arrow indicates the projection on the Multi-Dimensional Scaling (MDS) space of the number of geometric primitives in a shape. (**D**) The behavioral dissimilarity matrix (left) was better captured by a geometric feature coding model (middle) than by a convolutional neural network (right). The graph at right (**E**) shows the general linear model (GLM) coefficients for each participant. An accompanying explainer video is provided in *Figure 1—video 1*.

The online version of this article includes the following video, source data, and figure supplement(s) for figure 1:

**Figure supplement 1.** Additional convolutional neural network (CNN) encoding models of human behavior.

**Figure 1—video 1.** Explainer video for *Figure 1*.

https://elifesciences.org/articles/106464/figures#fig1video1

**Video 1—source data 1.** Explainer video transcript.

all simple, non-figurative, textureless four-sided shapes, yet varied in their geometric regularity. The most regular was the square, with four parallel sides of equal length and four identical right angles. By progressively removing some of these features (parallelism, right angles, equality of length, and equality of angles), we created a hierarchy of quadrilaterals ranging from highly regular to completely irregular (*Figure 1A*). In a variety of tasks, geometric regularity had a large effect on human behavior. For instance, for equal objective amounts of deviation, human adults and children detected a deviant shape more easily among shapes of high regularity, such as squares or rectangles (<5% errors), than among irregular quadrilaterals (>40% errors). The effect appeared as a human universal, present in preschoolers, first-graders, and adults without access to formal western math education (the Himba from Namibia), and thus seemingly independent of education and of the existence of linguistic labels for regular shapes. Strikingly, when baboons were trained to perform the same task, they showed no such geometric regularity effect.

Baboon behavior was accounted for by convolutional neural network (CNN) models of object recognition, but human behavior could only be explained by appealing to a representation of discrete geometric properties of parallelism, right angle, and symmetry, in this and other tasks. We sometimes refer to this model as 'symbolic' because it relies on discrete, exact, rule-based features rather than continuous representations (*Sablé-Meyer et al., 2022*). In this representational format, geometric shapes are postulated to be represented by symbolic expressions in a 'language-of-thought', for example 'a square is a four-sided figure with four equal sides and four right angles' or equivalently by a computer-like program from drawing them in a Logo-like language (*Sablé-Meyer et al., 2022*).

We therefore formulated the hypothesis that, in the domain of geometry, humans deploy an additional cognitive process specifically attuned to geometric regularities. On top of the circuits for object recognition, which are largely homologous in human and non-human primates (*Bao et al., 2020*; *Kriegeskorte et al., 2008b*; *Tsao et al., 2008*), the human code for geometric shapes would involve a distinct 'language of thought', an encoding of discrete mathematical regularities and their combinations (*Cavanagh, 2021*; *Dehaene et al., 2022*; *Fodor, 1975*; *Leeuwenberg, 1971*; *Quilty-Dunn et al., 2022*; *Sablé-Meyer et al., 2022*; *Sablé-Meyer et al., 2021*).

This hypothesis predicts that the most elementary geometric shapes, such as a square, are not solely processed within the ventral and dorsal visual pathways, but may also evoke a later stage of geometrical feature encoding in brain areas that were previously shown to encode arithmetic, geometric, and other mathematical properties, that is the bilateral intraparietal, inferotemporal, and dorsal prefrontal areas (*Amalric and Dehaene, 2016*; *Amalric and Dehaene, 2019*). We hypothesized that (1) such cognitive processes encode shapes according to their discrete geometric properties including parallelism, right angles, equal lengths, and equal angles; (2) the brain compresses this information when those properties are more regularly organized, and thus exhibit activity proportional to minimal description length (*Chater and Vitányi, 2003*; *Dehaene et al., 2022*; *Feldman, 2003*); and (3) these computations occur downstream of other visual processes, since they rely on the initial output of visual processing pathways.

Here, we assessed these spatiotemporal predictions using two complementary neuroimaging techniques (functional MRI and magnetoencephalography [MEG]). We presented the same 11 quadrilaterals as in our previous research and used representational similarity analysis (*Kriegeskorte et al., 2008a*) to contrast two models for their cerebral encoding, based either on classical CNN models or on exact geometric features. In the fMRI experiment, we also collected simpler images contrasting the category of geometric shapes to other classical categories such as faces, places, or tools. Furthermore, to evaluate how early the brain networks for geometric shape perception arise, we collected those fMRI data in two age groups: adults and children in first grade (6 years old, this year was selected as it marks the first year French students receive formal instruction in mathematics). If geometric shape perception involves elementary intuitions of geometric regularity common to all humans, then the corresponding brain networks should be detectable early on.

## Experiment 1: estimating the representational similarity of quadrilaterals with online behavior

Our research focuses on the 11 quadrilaterals previously used in our behavioral research (*Sablé-Meyer et al., 2021*), which are tightly matched in average pairwise distance between vertices and length of

the bottom side, yet differ broadly in geometric regularity (*Figure 1A*). The goal of our first experiment was to obtain an 11 × 11 matrix of behavioral dissimilarities between the 11 geometric shapes shown in *Figure 1A*, in order to compare it with predictions from classical visual models, embodied by CNNs, as well as geometric feature models of shape perception. To evaluate perceptual similarity in an objective manner, in experiment 1, we assessed the difficulty of visual search for one shape among rotated and scaled versions of the other (*Figure 1B*; *Agrawal et al., 2019*; *Agrawal et al., 2020*). Within a grid of nine shapes, eight are similar and one is different, and participants have to click on it. Intuitively, if two shapes are very dissimilar, we expect both the response time and the error rate of finding one exemplar of one shape among exemplars of the other shape to be low. Conversely, we expect both to be high if shapes are similar. This gives us an empirical measure of shape dissimilarity, which we can compare to the distance predicted by different models.

## Results

The 11 × 11 dissimilarity matrix, estimated by aggregating response time and errors from $n = 330$ online participants, is shown in *Figure 1D*. The distance was estimated as the average success rate divided by the average response time; method section for details. To better understand its similarity structure, we performed two-dimensional ordinal Multi-Dimensional Scaling (MDS) projection (*Figure 1C*; *De Leeuw and Mair, 2009*). The projection of the 11 shapes on the first dimension showed a strong geometric regularity, with the square and the rectangle landing at the far right, rhombus and parallelogram in the middle, and less regular shapes at the far left. Thus, human perceptual similarity seemed primarily driven by geometric properties. To quantify this resemblance using that 2D MDS projection, we examined the vector which corresponded to simply counting the number of geometric regularity (number of right angles, pairs of parallel lines, pairs of equal angles, and pairs of sides of equal length). This vector (shown in gray in *Figure 1C*) had a projection that was significantly different from 0 for both principal axes (both p < 0.01).

Most diagnostically, we compared the full human dissimilarity matrix to those generated by two competing models of shape processing (*Sablé-Meyer et al., 2021*). The geometric feature model proposes that each shape is encoded by a feature vector of its discrete geometric regularities and predicts dissimilarity by counting the number of features not in common: this makes squares and rectangles very similar, but squares and irregular quadrilaterals very dissimilar. On the other hand, we operationalize our visual model by propagating shapes through a feedforward CNN model of the ventral visual pathway (we use Cornet-S, but see *Figure 1—figure supplement 1* in Appendix for other CNNs and other layers of Cornet-S, with unchanged conclusions). Shape similarity was estimated as the cross-nobis distance between activation vectors in late layers (*Walther et al., 2016*). Note that these two models are not significantly correlated ($r^2 = 0.04$, p > 0.05).

Multiple regression of the human dissimilarity matrix with the predictions of those two visual and geometric feature models (*Figure 1E*) showed that both contributed to explaining perceived similarity (ps < 0.001), but that the weight of the geometric feature model was 3.6 times larger than that of the visual model, a significant difference (p < 0.001). This finding supports the prior proposal that the two strategies contribute to human behavior, but that the geometric feature-based one dominates, especially in educated adults (*Sablé-Meyer et al., 2021*). Additional models and comparisons are presented in *Figure 4—figure supplement 1*. In particular, we have included two distance measures based on skeletal representations (*Ayzenberg and Lourenco, 2019*; *Morfoisse and Izard, 2021*), both of which performed better than chance but significantly less than the geometric feature model. Finally, to separate the effect of name availability and geometric features on behavior, we replicated our analysis after removing the square, rectangle, trapezoids, rhombus, and parallelogram from our data (*Figure 1—figure supplement 1D*). This left us with five shapes and a representational dissimilarity matrix (RDM) with 10 entries. When regressing it in a general linear model (GLM) with our two models, we find that both models are still significant predictors (p < 0.001). The effect size of the geometric feature model is greatly reduced, yet remains significantly higher than that of the neural network model (p < 0.001).

# Experiment 2: fMRI geometric shape localizer

To understand the neural underpinning of the cognition of geometric shape using fMRI, we started with the simplest foray into geometric shape perception by including geometric shapes as an additional visual category in a standard visual localizer used in the lab (*Figure 2*). Across short miniblocks, this fMRI run probed whole-brain responses to geometric shapes (triangles, squares, hexagons, etc.) and to a variety of matched visual categories (faces, objects, houses, arithmetic, words, and Chinese characters). In 20 adults in functional MRI (9 females; 19–37 years old, mean age 24.6), we collected three localizer runs using a fast miniblock design. To maintain attention, participants were asked to detect a rare target, which could appear in any miniblock.

## Results

*Reduced activity to geometric shapes in ventral visual cortex.* Classical ventral visual category-specific responses, for instance to faces or words, were easily replicated (*Figure 2*; *Figure 2—figure supplement 1*). However, when contrasting geometric shapes to faces, houses, or tools, we observed a massive under-activation of bilateral ventral occipito-temporal areas (unless otherwise stated, all statistics are at voxelwise $p < 0.001$, clusterwise $p < 0.05$ permutation-test corrected for multiple comparisons across the whole brain). All of the regions specialized for words, faces, tools, or houses showed this activity reduction when presented with geometric shapes (*Figure 2C*; *Appendix 1—table 1*).

This group analysis was further supported by subject-specific analyses of regions of interest (ROIs) specialized for various categories of images (see Appendix and *Figure 2—figure supplement 3*). First, unsurprisingly, the fusiform face area, which is known for its strong category-selectivity, showed the lowest responses to geometric shapes in the fusiform face area. Second, in a subject-specific ROI analysis of the visual word form area (VWFA), identified by its stronger response to alphabetical stimuli than to face, tools, and houses, no activity was evoked by single shapes or strings of three shapes above the level of other image categories such as objects or faces. This finding eliminates the possibility that geometric shapes might have been processed similarly to letters. Similarly, one could have thought that geometric shapes would be processed together with other complex man-made objects, which are often designed to be regular and symmetrical. However, geometric shapes again yielded a low activation in individual ventral visual voxels selective to tools, thus refuting this possibility. Finally, geometric shapes could have been encoded in the parahippocampal place area (PPA), which is known to encode the geometry of scenes, including abstract ones presented as Lego blocks (*Epstein et al., 1999*). However, again, geometric shapes actually induced minimal activity in individually defined PPA voxels (see *Figure 2—figure supplement 3* for a summary).

*Increased activity to geometric shapes in intraparietal and inferior temporal cortices.* While the activity of the ventral visual cortex thus seemed to be globally reduced during geometric shape perception, we observed, conversely, a superior activation to geometric shapes than to face, tools, and houses in only two significant positive clusters, in the right anterior intraparietal sulcus (aIPS) and posterior inferior temporal gyrus (pITG) bordering on the lateral occipital sulcus. At a lower threshold (voxel $p < 0.001$ uncorrected), the symmetrical aIPS in the left hemisphere was also activated (also see Appendix for additional results concerning the '3-shapes' condition and with both shape conditions together).

Those areas are similar to those active during number perception, arithmetic, geometric sequences, and the processing of high-level math concepts (*Dehaene et al., 2022*; *Amalric and Dehaene, 2016*; *Amalric and Dehaene, 2019*; *Figure 2—figure supplement 4*). To test this idea formally, we used the localizer to identify ROIs activated by numbers more than words. This contrast identified a left IPS cluster ($p < 0.05$), while the symmetrically identified cluster in right IPS did not reach significance at the whole-brain level ($p = 0.18$). In both cases, however, the ROI was also significantly more activated by geometric shapes than other visual categories.

The observed overlap with number-related areas of the IPS is compatible with our hypothesis that geometric shapes are encoded as mental expressions that combine number, length, angle, and other geometric features. However, the association between geometric shapes and other arithmetic or mathematical properties could be acquired during schooling. To test whether the brain activity observed in adults reflects a basic intuition of geometry which is also present early on in child development, we replicated our fMRI study in 22 6-year-old first graders. When comparing the single shape

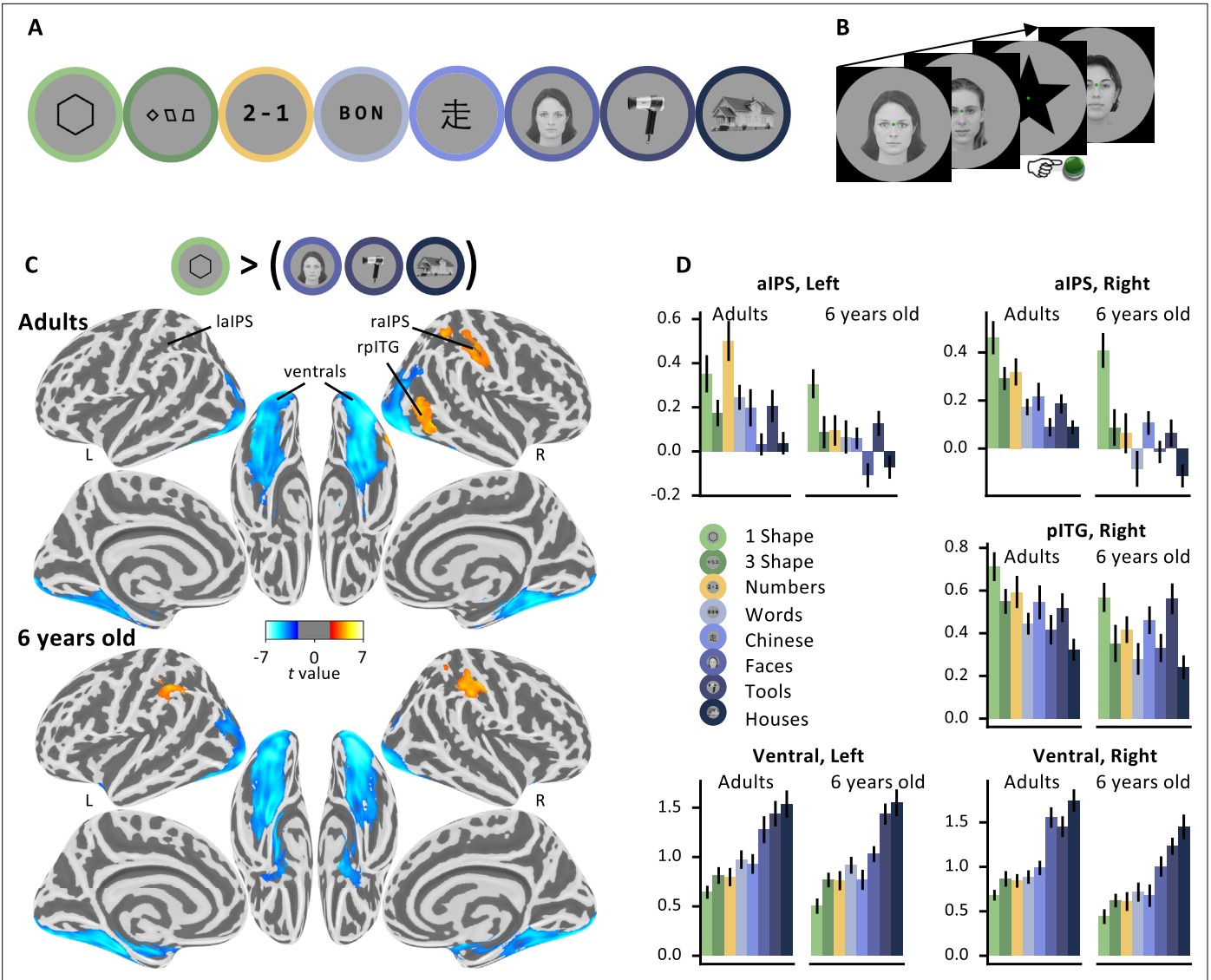

**Figure 2.** Localizing the brain systems involved in geometric shape perception. (**A**) Visual categories used in the localizer. (**B**) Task: Passive presentation by miniblocks of consistent visual categories. In some miniblock, among a series of six pictures from a given category, participants had to detect a rare target star. (**C**) Statistical map associated with the contrast 'single geometric shape > faces, houses, and tools', projected on an inflated brain (top: adults; bottom: children; clusters significant at cluster-corrected p < 0.05 with non-parametric two-tailed bootstrap test as reported in the text). (**D**) BOLD response amplitude (regression weights, arbitrary units) within each significant cluster with subject-specific localization. Geometric shapes activate the intraparietal sulcus (IPS) and posterior inferior temporal gyrus (pITG), while causing reduced activation in broad bilateral ventral areas compared to other stimuli; see *Figure 2—figure supplement 3* for analysis of subject-specific ventral subregions. An accompanying explainer video is provided in *Figure 2—video 1*.

The online version of this article includes the following video, source data, and figure supplement(s) for figure 2:

**Figure supplement 1.** Overview of the stimuli used for the category localizer.

**Figure supplement 2.** Details of the fMRI results in children.

(**A**) Statistical map associated with the contrast 'single geometric shape > faces, houses, and tools', projected on an inflated brain (top: adults; bottom: children; for illustration purposes, we display the uncorrected statistical map at the p < 0.01 level). Notice how similar the activations are in both age groups. (**B**) Same as A, but for the contrast 'single geometric shape > all single-object visual categories (face, house, tools, Chinese characters)'. The activation maps are very similar to the previous contrast and very similar across age groups. (**C**) Whole-brain correlation of the BOLD signal with geometric regularity in children, as measured by the error rate in a previous online intruder detection task (*Sablé-Meyer et al., 2021*). Positive correlations are shown in red and negative ones in blue. Voxel threshold p < 0.001, no correction for multiple comparisons, but the p-value indicates the only cluster that was significant at the cluster-level corrected p < 0.05 threshold. (**D**) Results of RSA analysis in children. No cluster was significant at the p < 0.05 level for the geometric feature models; one right-lateralized occipital cluster reached significance for the convolutional neural network (CNN)

*Figure 2 continued on next page*

*Figure 2 continued*

encoding model (cluster-level corrected p = 0.019), and its symmetrical counterpart was close to the significance threshold (cluster-level corrected p = 0.062).

**Figure supplement 3.** fMRI response of subject-specific voxels in the ventral visual pathway to geometric shapes and other visual stimuli.

**Figure supplement 4.** Overlap with math-responsive network and comparison with previous findings.

**Figure 2—video 1.** Explainer video for *Figure 2*.

https://elifesciences.org/articles/106464/figures#fig2video1

**Video 1—source data 1.** Explainer video transcript.

condition and faces, houses, and tools, we observed the same reduction in ventral visual activity. We also observed greater aIPS activity, now significant in both hemispheres, though the identified right pITG cluster did not reach significance. Still, the right pITG voxels extracted from adults reached significance in children for the shape versus face, houses, and tools. Conversely, the left aIPS voxels extracted in children reached significance in adults. Indeed, the activation profiles were quite similar in both age groups (*Figure 2D*; see *Figure 2—figure supplement 2* for uncorrected statistical maps of adults and children, which are quite similar). In particular, aIPS activity was strongest to geometric shapes in children, while in adults strong responses to both geometric shapes and numbers were seen, indicating an overlap with previously observed areas involved in arithmetic (*Amalric and Dehaene, 2019*; *Pinheiro-Chagas et al., 2018*) (see *Figure 2—figure supplement 1*).

To better establish the correspondence between our findings and previous work, we evaluated the 'geometric shape > other visual categories' contrast in six ROIs previously identified as part of the math-responsive network (*Amalric and Dehaene, 2016*): bilateral IPS, bilateral MFG, and bilateral pITG. In all ROIs, in both adults and children, the average geometric shape contrast was significant at the $p < 0.05$ level, except for the left posterior ITG in children ($p = 0.09$). Nevertheless, cautiousness is required here because activation overlap could arise without indicating that the same exact circuits and processes are involved, especially in smoothed group-level images. To partially mitigate this problem and test for subject-level overlap between the activations to geometric shapes and to numbers, we turned to within-subject analyses. We assessed whether the patterns of activation evoked by numbers and geometric shapes were more similar to each other than those evoked by numbers and other categories. For each subject, we computed the cross-nobis distance between the activations evoked by each of the experimental conditions. We then compared the distances for numbers versus geometric shapes, and for numbers versus the average of the other categories. We found that, in adults, in four of our five ROIs, the activations to geometric shapes were indeed more similar to numbers than to other categories ($p < 0.05$ in lIPS, rITG, and bilateral ventral ROIs, $p = 0.17$ in rIPS). In children, the pattern of similarity was too noisy to be conclusive for bilateral IPS or the ITG, but the finding remained significant in bilateral ventral areas (both $p < 0.05$).

In sum, geometric shapes led to reduced activity in ventral visual cortex, where other categories of visual images show strong category-specific activity. Instead, geometric shapes activated areas independently found to be activated by math- and number-related tasks, in particular the right aIPS.

## Experiment 3: fMRI of the geometric intruder task

In a second fMRI experiment, we measured the detailed fMRI patterns evoked by our quadrilaterals, with the goal to submit them to a representational similarity analysis and test our hypothesis of a double dissociation between regions encoding visual (CNN) versus geometric codes. Adults and children performed an intruder task similar to our previous behavioral study (*Sablé-Meyer et al., 2021*), with miniblocks allowing us to evaluate the activity pattern evoked by each quadrilateral shape. To render the task doable by 6-year-olds, we tested only 6 quadrilaterals, displayed as two half-circles of three items, and merely asked participants whether the intruder was on the left or the right of the screen (*Figure 3*).

### Results

Behavioral performance inside the scanner replicated the geometric regularity effect in adults and children: performance was best for squares and rectangles and decreased linearly for figures with

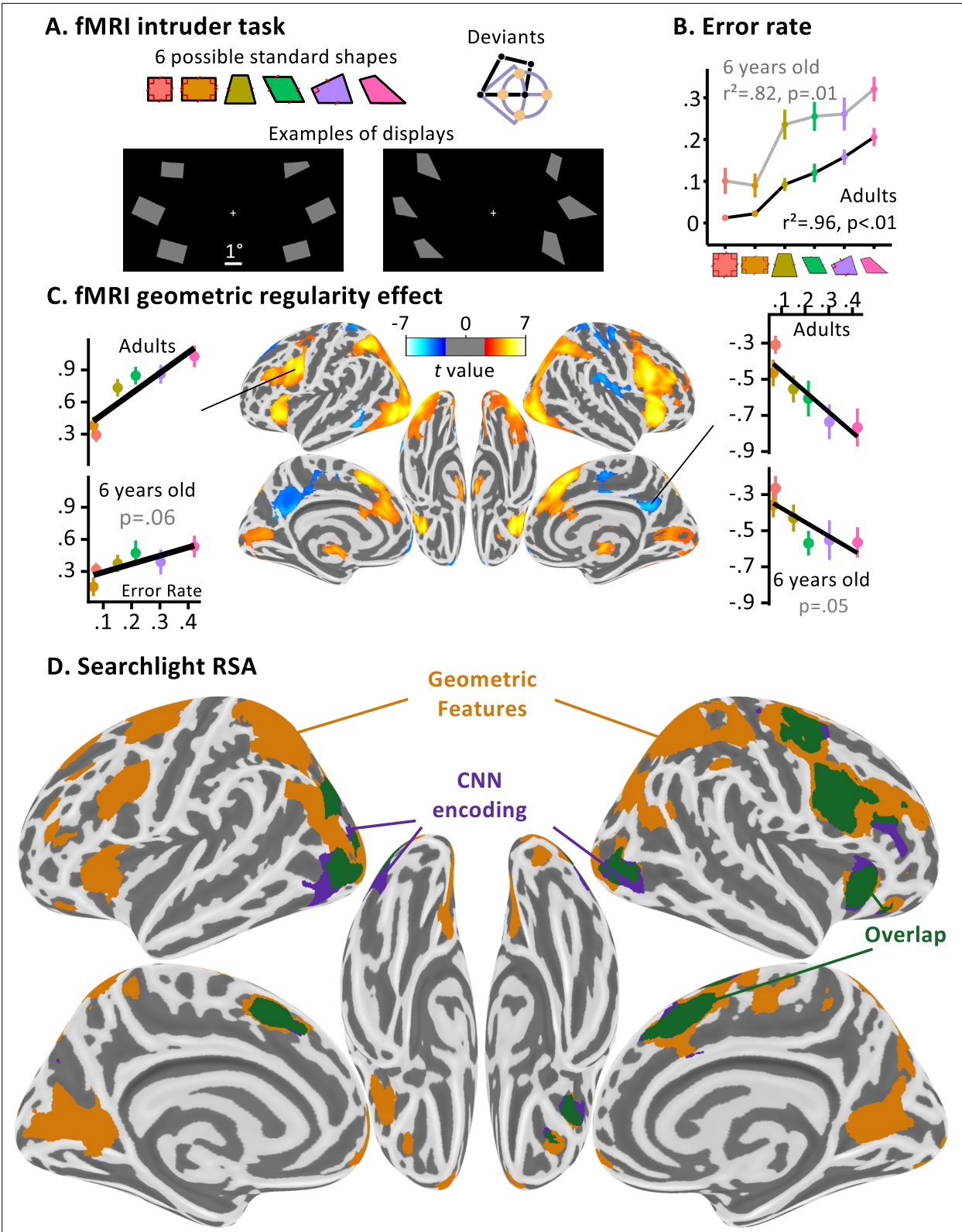

**Figure 3.** Dissociating two neural pathways for the perception of geometric shape. (**A**) fMRI intruder task. Participants indicated the location of a deviant shape via button clicks (left or right). Deviants were generated by moving a corner by a fixed amount in four different directions. (**B**) Performance inside the fMRI: both populations tested displayed an increasing error rate with geometric shape complexity, which significantly correlates with previous data collected online. (**C**) Whole-brain correlation of the BOLD signal with geometric regularity in adults, as measured by the error rate in a previous

*Figure 3 continued on next page*

*Figure 3 continued*

online intruder detection task (*Sablé-Meyer et al., 2021*). Positive correlations are shown in red and negative ones in blue. Voxel threshold p < 0.001, cluster-corrected by permutation at p < 0.05. Side panels show the activation in two significant regions of interest (ROIs) whose coordinates were identified in adults and where the correlation was also found in children (one-tailed test, corrected for the number of ROIs tested this way). (**D**) Whole-brain searchlight-based RSA analysis in adults (same statistical thresholds). Colors indicate the model which elicited the cluster: purple for convolutional neural network (CNN) encoding, orange for the geometric feature model, green for their overlap. An accompanying explainer video is provided in *Figure 3—video 1*.

The online version of this article includes the following video, source data, and figure supplement(s) for figure 3:

**Figure 3—video 1.** Explainer video for *Figure 3*.

https://elifesciences.org/articles/106464/figures#fig3video1

**Video 1—source data 1.** Explainer video transcript.

**Figure supplement 1.** Additional models: fMRI.

fewer geometric regularities (*Figure 3B*). Behavior in the fMRI scanner was significantly correlated with an empirical measure of geometric regularity measured with an online intruder detection task (*Sablé-Meyer et al., 2021*), which is used in all following analyses whenever we refer to geometric regularity. The neural bases of this effect were studied at the whole-brain level by searching for areas whose activation varied monotonically with geometrical regularity. In adults, this contrast identified a broad bilateral dorsal network including occipito-parietal, middle frontal, anterior insula, and anterior cingulate cortices, with accompanying deactivation in posterior cingulate (*Figure 3C*). This broad network, however, probably encompassed both regions involved in geometric shape encoding and those involved in the decision about the intruder task, whose difficulty increased when geometric regularity decreased. To isolate the regions specifically involved in shape coding, we performed searchlight RSA analyses, which focused on the pattern rather than the level of activation. Within spheres spanning the entire cortex, we asked in which regions the similarity matrix between the activation patterns evoked by the six shapes was predicted by the CNN encoding model, the geometric feature model, or both (*Figure 3*; *Appendix 1—table 2*). In adults, the CNN encoding model predicted neural similarity in bilateral lateral occipital clusters, while the geometric feature model yielded a large set of clusters in occipito-parietal, superior prefrontal, and right dorsolateral prefrontal cortex. The calcarine cortex was also engaged, possibly due to top–down feedback (*Williams et al., 2008*). Both CNN encoding and geometric feature models had overlapping voxels in anterior cingulate and right premotor cortex, possibly reflecting the pooling of both visual codes for decision-making.

In children, possibly because the task difficulty was high, few results were obtained. The correlation of brain activity with regularity at the whole-brain level yielded a single significant cluster (see *Figure 2—figure supplement 2* for an uncorrected statistical map (voxelwise p < 0.001), with the single significant whole-brain, cluster corrected significant cluster (p = 0.012) in the ventromedial prefrontal cortex). When testing the ROIs from adults in children, after correcting for multiple comparisons across the 14 ROIs, only the posterior cingulate was significant (p = 0.049), though two clusters were close to significance, both with p = 0.06: one positive in the left precentral gyrus (shown in *Figure 3C*), and one negative in the dorsolateral prefrontal cortex. Testing the ROIs identified in the visual localizer showed that they all exhibited a positive geometric difficulty effect in adults (all p < 0.05), but did not reach significance in children, possibly due to excessive noise (as indicated by the much higher error rate in the task inside the scanner). In the searchlight analysis, no cluster associated with the geometric feature model reached significance at the whole-brain level (*Figure 2—figure supplement 2*). However, a right lateral occipital cluster was significantly captured by the CNN encoding model in children (p = 0.019) and its symmetrical counterpart was close to the significance threshold (p = 0.062) (*Figure 2—figure supplement 2*). This result might indicate that geometric features are not well differentiated prior to schooling. It could also reflect that children weight the geometric feature strategy less, as was found in previous work (*Sablé-Meyer et al., 2021*); combined with difficulty of obtaining precise subject-specific activation patterns in young children, this could make the geometric feature strategy harder to localize.

## Experiment 4: oddball paradigm of geometric shapes in MEG

The temporal resolution of fMRI does not allow tracking the dynamic of mental representations over time. Furthermore, the previous fMRI experiment suffered from several limitations. First, we studied six quadrilaterals only, compared to 11 in our previous behavioral work (*Sablé-Meyer et al., 2021*). Second, we used an explicit intruder detection, which implies that the geometric regularity effect was correlated with task difficulty, and we cannot exclude that this factor alone explains some of the activations in *Figure 3C* (although it is much less clear how task difficulty alone would explain the RSA results in *Figure 3D*). Third, the long display duration, which was necessary for good task performance especially in children, afforded the possibility of eye movements, which were not monitored inside the 3T scanner and again could have affected the activations in *Figure 3C*.

To overcome those issues, we replicated the experiment in adult MEG with three important changes: (1) all 11 quadrilaterals were studied; (2) participants were simply asked to fixate and attend to every shape, without performing any explicit task; (3) shapes were presented serially, one at a time, at the center of screen, with small random changes in rotation and scaling parameters; and (4) in miniblocks of 30 s each, a fixed quadrilateral shape appeared repeatedly, interspersed with rare intruders whose bottom right corner was shifted by a fixed amount (*Sablé-Meyer et al., 2021*). This design allowed us to study the neural mechanisms of the geometric regularity effect without confounding effects of task, task difficulty, or eye movements. Would the shapes be automatically encoded according to their geometric regularities even in such a passive context? And would brain responses indicate that the intruders continued to be detected more easily among geometric regular shapes than among irregular ones, as previously found behaviorally in an active task (*Sablé-Meyer et al., 2021*)?

### Results

In spite of the passive design, MEG signals revealed an automatic detection of intruders, driven by geometric regularity. We trained logistic regression decoders to classify the MEG signals at each timepoint following a shape as arising from a reference shape or from an intruder (see Method and *Figure 4A*). Overall, the decoder performed above chance level, reaching a peak at 428 ms, indicating the presence of brain responses specific to intruders. Crucially, although trained on all shapes together, the decoder performed better with geometrically regular shapes than with irregular shapes (and better than chance for each), indicating that oddball shapes were more easily detected within blocks of regular shapes, as previously found behaviorally (*Sablé-Meyer et al., 2021*). Indeed, a regression of decoding performance on geometrical regularity yielded a significant spatiotemporal cluster of activity, which first became significant around ~160 ms and peaked at 432 ms (here and after, temporal clusters are purposefully reported with approximate bounds following *Sassenhagen and Draschkow, 2019*). A geometrical regularity effect was also seen in the latency of the outlier response (see *Figure 4A*; correlation between geometric regularity and the latency when average decoding performance first exceeded 57% correct; one-tailed *t*-test of each participant's regression slope against 0: $t = -1.83$, $p = 0.041$) indicating that oddballs yielded both a larger and a faster response when the shapes were geometrically more regular. The same effect was found when training separate outlier decoders for each shape, thus refuting an alternative hypothesis according to which all outliers evoke identical amounts of surprise, but in different directions of neural space (*Figure 4B*). Overall, the results fully replicate prior behavioral observations of the geometric regularity effect (*Sablé-Meyer et al., 2021*) and suggest that the computation of a geometric code and its deviations occurs under passive instructions and starts with a latency of about ~150 ms.

We next used temporal RSA to further probe the dynamics of the perception of the reference, non-intruder shapes. For each timepoint, we estimated the 11 × 11 neural dissimilarity matrix across all pairs of reference shapes using sensor data with cross-nobis distances (*Walther et al., 2016*), and entered them in a multiple regression with those predicted by CNN encoding and geometric feature models. The coefficients associated with each predictor are shown in *Figure 4C*. An early cluster (observed cluster extent at approximately 60–320 ms, peak at 84 ms; $p < 0.001$) showed a significant correlation of the CNN encoding model on brain similarity. It was followed by two significant clusters associated with the geometric feature model separated by two timepoints that did not pass the cluster-formation threshold (~128–184 ms then ~196–400 ms, overall peak at 232 ms; $p < 0.001$). Those results are compatible with our hypothesis of two distinct stages in geometric shape perception and suggest that a geometric feature-based encoding is present, especially around ~200–250

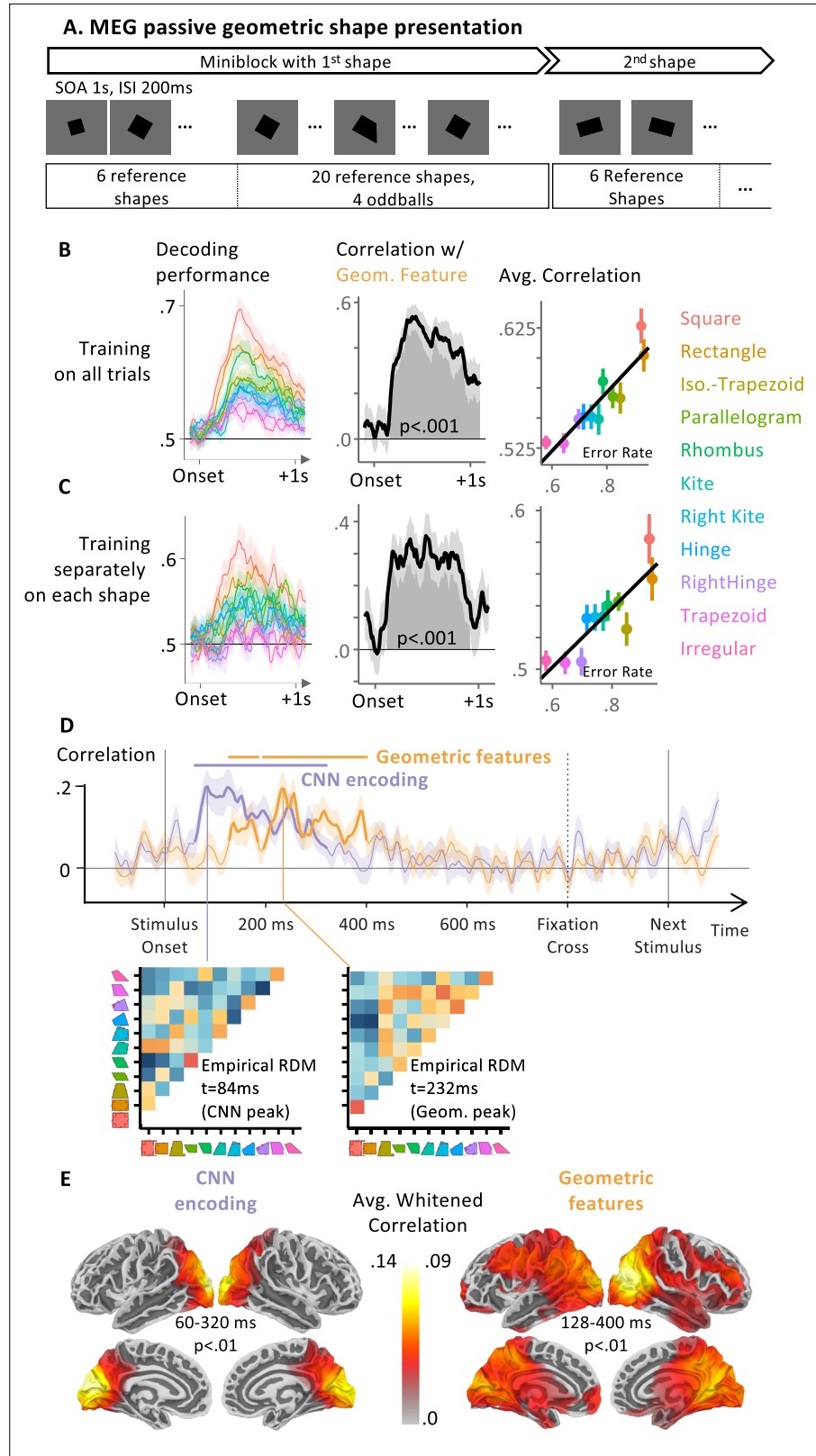

**Figure 4.** Using magnetoencephalography (MEG) to time the two successive neural codes for geometric shapes. (**A**) Task structure: participants passively watch a constant stream of geometric shapes, one per second (presentation time 800 ms). The stimuli are presented in blocks of 30 identical shapes up to scaling and rotation, with four occasional deviant shapes. Participants do not have a task to perform besides fixating. (**B, C**) Performance

*Figure 4 continued on next page*

*Figure 4 continued*

of a classifier using MEG signals to predict whether the stimulus is a regular shape or an oddball. Left: performance for each shape; middle: correlation with geometrical regularity (same x axis as in *Figure 3C*); right: visualization of the average decoding performance over the cluster. In B, training of the classifier was performed on MEG signals from all 11 shapes; In C, 11 different classifiers were trained separately, one for each shape. (**D**) Sensor-level temporal RSA analysis. At each timepoint, the 11 × 11 dissimilarity matrix of MEG signals was modeled by the two model representational dissimilarity matrices (RDMs) in *Figure 1D*, and the graph shows the time course of the corresponding whitened correlation coefficients. Below the time courses, we display the average empirical dissimilarity matrix across participants at two notable timepoints: when the correlation with the convolutional neural network (CNN) and geometric feature models are maximal (CNN: *t* = 84 ms; geometric features: 232 ms) (**E**) Source-level temporal-spatial searchlight RSA. Same analysis as in C, but now after reconstruction of cortical source activity. An accompanying explainer video is provided in *Figure 4—video 1*.

The online version of this article includes the following video, source data, and figure supplement(s) for figure 4:

**Figure 4—video 1.** Explainer video for *Figure 4*.

https://elifesciences.org/articles/106464/figures#fig4video1

**Video 1—source data 1.**

**Figure supplement 1.** Additional models: behavior and magnetoencephalography (MEG).

---

ms. This analysis yielded similar clusters when performed on a subset of shapes that do not have an obvious name in English, as was the case for the behavior analysis (CNN encoding: 64.0–172.0 ms; then 192.0–296.0 ms; both p < 0.001: geometric features: 312.0–364.0 ms with p = 0.008, the later timing could indicate that geometric features for less regular shapes take longer to estimate, replicating the latency effect found in the oddball decoding analysis).

To understand which brain areas generated these distinct patterns of activations and probe whether they fit with our previous fMRI results, we performed a source reconstruction of our data. We projected the sensor activity onto each participant's cortical surfaces estimated from T1 images. The projection was performed using eLORETA (*Jatoi et al., 2014*) and empty room recordings acquired on the same day to estimate noise covariance, with the default parameters of mne-bids-pipeline. Sources were spaced using a recursively subdivided octahedron (oct5). Group statistics were performed after alignment to fsaverage. We then replicated the RSA analysis with searchlights sweeping across cortical patches of 2 cm geodesic radius (*Figure 4D*). The CNN encoding model captured brain activity in a bilateral occipital and posterior parietal cluster, while the geometric model accounted for subsequent activity starting at ~200 ms and spanning over broader dorsal occipito-parietal and intraparietal, prefrontal, anterior, and posterior cingulate cortices. This double dissociation closely paralleled fMRI results (compare *Figures 3D and 4D*; see also *Video 1* for a movie of the significant sources associated to either model across time), with greater spatial spread due to the unavoidable imprecision of MEG source reconstruction.

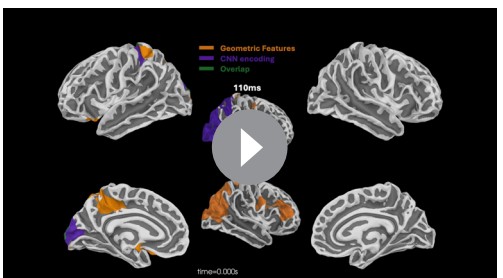

**Video 1.** Spatiotemporal dynamics of the magnetoencephalography (MEG) data compared to the models. Searchlight-based timepoint-per-timepoint RSA analysis across shapes of the MEG data. Significant (p < 0.05) sources associated with the convolutional neural network (CNN) model are shown in purple, significant sources associated with the geometric feature model in orange; overlap is shown in green. Cluster-corrected clusters reported in *Figure 4*.

https://elifesciences.org/articles/106464/figures#video1

## Modeling of the results with alternative models of visual perception

In the above analyses, we contrasted two models for our data: a CNN versus a list of geometric properties. However, there are many other models of vision. In this section, we model our data with two additional classes of models, based on the shape skeleton and principal axis theory, or vision transformers neural networks.

### Skeleton models

Skeletal representations (*Blum, 1973*) offer biologically plausible and computationally

tractable models of object representation in human cognition. Several methods to derive skeletal representation from object contour have been put forward, including Bayesian estimation that trades off accuracy for conciseness (*Feldman and Singh, 2006*). More recently, distance estimators between shapes based on their skeletal structure have been used to model human behavior (*Ayzenberg et al., 2022*; *Ayzenberg and Lourenco, 2019*; *Denisova et al., 2016*; *Lowet et al., 2018*; *Morfoisse and Izard, 2021*), and hierarchical object decomposition has been rigorously described and tested (*Froyen et al., 2015*). Remarkably, even when simply asked to tap a geometric shape on a touchscreen anywhere they want, human taps cluster along the shape skeleton (*Firestone and Scholl, 2014*).

To model our data, we needed a distance metric between two shape skeletons. We explored two metrics: one that measures the shortest distance between two skeletons after optimal alignment and rotation (*Ayzenberg and Lourenco, 2019*; *Jacob et al., 2021*), and one that measures the difference in angle and length of matched sub-parts of the shapes' skeleton (*Morfoisse and Izard, 2021*). Our results are summarized in *Figure 3—figure supplement 1A*; the first observation is that over our 11 quadrilaterals, these two skeletal metrics were not significantly correlated, suggesting that they capture very different properties when applied to minimal geometric shapes and are possibly not best suited to characterize them. Additionally, neither of these metrics predicted the behavioral data better than the CNN models.

When tested on fMRI data, the first method based on optimal alignment and rotation did not elicit any significant cluster in either population. In adults, but not in children, the second method based on the structure of the skeletons significantly correlated with bilateral clusters in the visual cortex, as well as a significant cluster in the right premotor cortex (see *Figure 3—figure supplement 1*). These areas constitute a subset of the areas identified with the CNN encoding model.

When tested on MEG data, the first method similarly did not elicit any significant temporal cluster at the p < 0.05 level. The second method yielded two significant temporal clusters, the first from ~75 to ~260 ms and the second from ~285 to ~370 ms. When looking at the reconstructed sources, the second temporal cluster did not yield any significant spatial cluster, and the first cluster elicited a bilateral cluster encompassing both early occipital area and the early dorsal visual stream, with no significant localization in the frontal cortex.

Overall, for our dataset, these models appear partially similar to the CNN model insofar as they capture subsets of the timings and localizations of the neural data captured by the CNN model, but they do not strongly predict the behavior data as the geometric feature model does, nor capture the broad cortical networks associated with the geometric feature model.

## Vision transformers and larger neural networks

While small CNN models have well-known limits in capturing several aspects of human perception (*Bowers et al., 2022*; *Jacob et al., 2021*), those limits are constantly being pushed. First, connectionist models based on the more advanced transformer architecture may provide a closer match to human data, especially when it comes to symbolic or mathematical regularities (*Campbell et al., 2024*). Second, even within the CNN-like architecture, much larger neural networks, trained on vastly richer datasets, may achieve superior similarity to humans than CORnet. We briefly consider both possibilities.

First, using the same method as with CNNs, we modeled our empirical RDMs with distance measured between pairs of shapes in the late layers of a visual transformer, DINOv2 (*Oquab et al., 2023*). In *Figure 4—figure supplement 1B*, we report the empirical RDMs, the DINO RDM, as well as the CORnet CNN RDM, together with a plot of each subject's correlation with the CNN and DINO matrices. The DINO predictor yielded a dissimilarity matrix quite similar to the human behavioral one; indeed, the distribution of coefficients for the DINO predictor was significantly greater than 0 (95% confidence interval [0.73, 0.75]; p < 0.001) and significantly different from the CNN predictor (p < 0.001). Note that in this analysis, the distribution of CNN predictors is significantly lower than zero (p < 0.001), which was not the case when contrasted with the symbolic model. This suggests that the DINO predictor may involve a mixture of symbolic-like features and perceptual features. Indeed, its correlation with the symbolic model is 0.78 and the correlation with the cornet is 0.56. This may explain why humans are best predicted with a mixture of these two models, but with different weights: to achieve the best fit, a model with both DINO and CNN puts a high weight on DINO and then negatively corrects the excess perceptual features by putting a negative weight on the CNN model.

This analysis was confirmed by a fit of the fMRI data (*Figure 3—figure supplement 1*) and MEG data (*Figure 4—figure supplement 1C*): in both cases, the RSA analysis with DINO last layer yielded results that were very similar to a mixture of the CORnet CNN and the symbolic model. In fMRI in adults, a distributed set of cortical areas was significantly correlated with the DINO model, and these areas overlapped with the union of the areas predicted by both of our original models. In children, DINOv2 elicited a significant cluster in the right visual cortex, in areas overlapping with the areas activated by the CNN model in children.

In MEG, a large and significant cluster was predicted by the DINO model. Its timing (significant cluster from ~64 to ~400 ms) overlapped with the spatiotemporal clusters found with both the CNN model and the symbolic model. Source analysis indicated that a correlation with DINO originated from widespread sources that, again, overlapped with both those found with the CNN model (early occipital areas) and the symbolic model (widespread regions included dorsal, anterior temporal, and frontal sources).

Taken together, these results suggest that the last layer of DINO is driven by both early visual and higher-level geometric features, and that both are needed to fit human data. While these findings may suggest that network architecture may be crucial to mimic human geometric perception, continuing work in our lab challenges this conclusion because similar results were obtained by performing the same analysis, not only with another vision transformer network, ViT, but crucially using a much larger convolutional neural network, ConvNeXT, which comprises ~800 million parameters and has been trained on two billion images, likely including many geometric shapes and human drawings. For the sake of completeness, RSA analysis in sensor space of the MEG data with these two models is provided in *Figure 4—figure supplement 1*. A systematic investigation of the impact of network architecture, dataset size, and dataset content is beyond the scope of the present paper, but the present data could serve as a reference dataset with which to understand which of these factors ultimately cause a neural network to display symbolic properties close to those found in the human brain.

## Discussion

Our previous behavioral work suggested that the human perception of geometric shapes cannot be solely explained by simple CNN encoding models – rather, humans also encode them using nested combinations of discrete geometric regularities (*Sablé-Meyer et al., 2022*; *Dehaene et al., 2022*; *Sablé-Meyer et al., 2021*). Here, we provide direct human brain-imaging evidence for the existence of two distinct cortical networks with distinct representational schemes, timing, and localization: an early occipitotemporal network well modeled by a simple CNN, and a dorso-frontal network sensitive to geometric regularity.

### Behavioral signatures of geometric regularity

At the behavioral level, the new large-scale data that we report here (*n* = 330 participants) fully replicated our prior finding that human perception of geometric shape dissimilarity, as evaluated by a visual search task, is best predicted by a model that relies on exact geometric features, rather than by a CNN. These exact geometric feature representations can be thought of as a more abstract and compressed representations of shapes: they replace continuous variations along certain dimensions (such as angles or directions) with categorical features (right angle or not, parallel or not). Such a representation, which remains invariant over changes in size and planar rotation, turns a rich visual stimulus into a compressed internal representation.

Even in educated human adults, a small proportion of the variance in behavioral dissimilarity remained predicted by the simple CNN model, so that both CNN and geometry predictors were significant in a multiple regression of human judgments. Previously, we only found such a significant mixture of predictors in humans without formal western education (whether French preschoolers or adults from the Himba community, mitigating the possible impact of explicit western education, linguistic labels, and statistics of the environment on geometric shape representation) (*Sablé-Meyer et al., 2021*). It is likely that the greater power afforded by the present experiment yielded a greater sensitivity. This finding comforts the hypothesis that two strategies for geometric shape perception are available to humans: one based on hierarchical visual processing (captured by the CNN), and the

other based on an analysis of discrete geometric features – the latter dominating strongly in educated adults.

## fMRI results

Using fMRI, we found that the mere perception of geometric shapes, compared to various other visual categories, yields a reduced bilateral activation of the entire ventral visual pathway, together with a localized increased activation in the anterior IPS and in the posterior ITG. Furthermore, geometric regularity modulated fMRI activation in most of the bilateral occipito-parietal pathway, middle frontal gyrus, and anterior insula. Finally, the most diagnostic result we found is that the representational similarity matrix in these regions was predicted by geometric similarity rather than by the simple CNN (*Figure 3*).

The IPS areas activated by geometric shapes overlap with those active during the comprehension of elementary as well as advanced mathematical concepts (*Dehaene et al., 2022*; *Amalric and Dehaene, 2016*; *Amalric and Dehaene, 2019*). Although this finding must be interpreted with great caution, as activation overlap need not imply shared cognitive processes, it was confirmed at the single-subject level and agrees with the proposed involvement of these regions in an abstract, modality-independent symbolic encoding of shapes. Further support for this idea comes from the fact that these regions can be equally activated in sighted and in blind individuals when they perform mathematics (*Amalric et al., 2018*; *Kanjlia et al., 2016*) or when they evaluate the shapes of manmade objects (*Xu et al., 2023*). Thus, the representation of shapes computed in these regions is more abstract and amodal than the hierarchy of visual filters that is thought to capture ventral visual image recognition.

One referee noted that we contrasted geometric shapes with other categories of images that may, themselves, possess some geometric features. For instance, faces possess a plane of quasi-symmetry, and so do many other man-made tools and houses. Thus, our subtraction isolated the geometrical features that are present in simple regular geometric shapes (e.g. parallels, right angles, and equality of length) over and above those that might already exist, in a less pure form, in other categories.

## MEG results

Using MEG, we found that even during the passive perception of simple quadrilateral shapes, in the absence of an explicit task, participants were sensitive to their geometric regularity: occasional oddball shapes elicited greater violation-of-expectation signals within blocks of regular shapes than within blocks of irregular shapes. Additionally, using RSA and source reconstruction, we observed a spatial and temporal dissociation in the processing of quadrilateral shapes: an early occipital response, well predicted by CNN models of object recognition, was followed by broadly distributed cortical activity that correlated with a model of exact geometric features. Despite the limited accuracy afforded by source reconstruction in MEG, there was considerable overlap between our RSA analysis in fMRI and in MEG.

This early response of occipital areas, followed by a later activation of a broad dorso-frontal network, may seem at odds with some results showing that, during visual image processing, the dorsal pathway may respond faster than the ventral pathway (*Ayzenberg et al., 2023*; *Collins et al., 2019*). However, in this work, we specifically probed the processing of geometric shapes that, if our hypothesis is correct, are represented as mental expressions that combine geometrical and arithmetic features of an abstract categorical nature, for instance representing 'four equal sides' or 'four right angles'. It seems logical that such expressions, combining number, angle, and length information, take more time to be computed than the first wave of feedforward processing within the occipito-temporal visual pathway, and therefore only activate thereafter.

## Reduced activation of the ventral visual pathway

Strikingly, our fMRI data also evidenced a hypo-activation of the ventral visual pathways to geometric shapes relative to other pictures. A large bilateral cluster of reduced activity was found when comparing shapes versus other visual categories, and a similar reduced activity was found in each of four visual areas specialized for various visual categories (faces, words, tools, and places). This reduction is compatible with our hypothesis that geometric shapes are processed differently from other pictures, and with empirical finding that a simple feedforward CNN, whose architecture usually provides a relatively good fit to inferotemporal cortex activity (*Conwell et al., 2021*; *Kubilius et al.,*

*2019*; *Schrimpf et al., 2018*; *Yamins et al., 2014*), did not provide a good model of geometric shape perception. Because our shapes consisted of a few straight lines, it is likely that they solicited the computations performed by the ventral pathway only minimally, giving rise to a lower activation when compared to more complex visual stimuli such as pictures of faces or houses.

## Two visual pathways

Brain-imaging studies of the ventral visual pathway indicate that it is subdivided into patches responding to different visual categories such as faces, tools, or houses, with a partial correspondence between human and non-human primates (*Arcaro et al., 2017*; *Margalit et al., 2020*; *Rauschecker et al., 2012*; *Tsao and Livingstone, 2008*; *Khosla et al., 2022*; *Allen et al., 2021*; *Kriegeskorte et al., 2008b*). Converging evidence from behavior (*Biederman and Ju, 1988*) and neuroimaging (*Ayzenberg et al., 2022*; *Lescroart and Biederman, 2013*; *Papale et al., 2019*; *Papale et al., 2020*) has underlined the central role of the shape of objects, contour, and texture in object recognition. A basic complementarity has been identified between the ventral visual pathway, crucial for visual identification, and the dorsal occipito-parietal route, extracting visuo-spatial information about orientation and motor affordances. However, this simplified view needs to be nuanced by evidence of shape processing in the dorsal pathway as well, especially in posterior areas (*Freud et al., 2016*; *Xu, 2018*), in both human and non-human primates. More specifically, recent work challenges the idea that the shape of objects is computed solely in the ventral pathway, and instead suggests that the dorsal pathway, and in particular the IPS, may first compute global shape information and then propagate it to the ventral pathway for further processing (*Ayzenberg and Behrmann, 2022*). Our results are compatible with this new look at dorsal/ventral interactions: in our fMRI localizer, geometric shapes elicited reduced ventral and increased dorsal activation when compared to other visual categories. Geometric shapes could be considered as conveying very pure information about global shape, entirely driven by contour geometry and fully devoid of other information such as texture, shading, or curvature. More research is needed to understand the dynamic collaboration between the ventral and dorsal streams during shape recognition, but the present results, as well as the arguments put forward in *Ayzenberg and Behrmann, 2022*, concur to suggest that shape identification does not solely rely on the ventral visual pathway.

Interestingly, recent work shows that within the visual cortex, the strongest relative difference in cortical surface expansion between human and non-human primates is localized in parietal areas (*Meyer et al., 2025*). If this expansion reflected the acquisition of new processing abilities in these regions, it might explain the observed differences in geometric abilities between human and non-human primates (*Sablé-Meyer et al., 2021*).

## Lateral occipital cortex

Of special relevance to this work is the role of the lateral occipital cortex (LOC) in shape perception (*Grill-Spector et al., 2001*). The posterior ITG activation that we observed lies just anterior to the LOC and possibly overlapped with it. The LOC has been repeatedly associated with shape processing (*Grill-Spector et al., 1998*; *Kanwisher et al., 1996*; *Kourtzi and Kanwisher, 2000*; *Malach et al., 1995*), and our work converges to suggest that this region and the cortex just anterior to it play a key role in the perception of simple shapes in a way that is invariant to 2D and 3D rotations (*Kourtzi et al., 2003*). However, unlike ours, previous work focused primarily on irregular potato-like shapes or objects and their contours.

Object selectivity in the LOC is already present in early infancy, with a sensitivity to shape and not texture already observed at 6 months of age (*Emberson et al., 2017*). At age 5–10, size invariance, but not viewpoint invariance, has been established (*Nishimura et al., 2015*). However, to our knowledge, no study has targeted geometrically regular shapes, and in particular how changes in regularity impact LOC activity. While some of our quadrilateral stimuli can be constructed as 3D projections of one another (either by perspective or by orthogonal projection), this should make them more similar to each other within the LOC given its object viewpoint invariance. Further work should use within-subject LOC localizers such as objects minus scrambled-objects to establish the exact cortical relation between the present geometric regularity effect and the classical LOC region. We speculate that the more anterior part of lateral occipito-temporal cortex may extract more abstract geometric

descriptors of shapes than the classical LOC, as also suggested by its sensitivity to mathematical training (*Amalric and Dehaene, 2016*).

## Development of geometry representations

Interestingly, the IPS and posterior ITG activations to geometric shapes were consistently observed at very similar locations in adult and 6-year-olds. Furthermore, the overlap with a math-responsive network was present in both age groups, with ROIs that respond to number more than words also responding to geometric shapes more than other visual categories. This finding fits with previous evidence of an early responsivity of the IPS to numbers and arithmetic in young children prior to formal schooling (*Ansari, 2008*; *Cantlon and Li, 2013*; *Izard et al., 2008*). In agreement with the existence of a behavioral geometric regularity effect in preschoolers (*Sablé-Meyer et al., 2021*), we hypothesize that an intuitive encoding of number and geometric features precedes schooling and possibly guides the subsequent acquisition of formal mathematics (*Izard et al., 2022*).

On the other hand, the fMRI results from the intruder task were much less stable in children, and neither the geometric regularity effect nor the geometry-based RSA analysis yielded strong results at the whole-brain level (only one ventromedial cluster was associated with the regularity effect, see *Figure 2—figure supplement 2*). In previous work, we have shown that besides an overall decrease in performance between adults and first graders, the profile of behavior across different shapes is partially distinct in preschoolers: in them, both the CNN and the symbolic model are on par in predicting behavior, suggesting that unlike adults, children frequently mix the two strategies for identifying geometric shapes. Such a mixture of strategies could explain why the correlation with our empirical estimation of complexity, based on adult data, yielded a weaker and more noisy correlation in children. The RSA analysis is compatible with this: in the group analysis, while the CNN predictor was significant in the visual cortex of children, the symbolic geometry model did not reach significance anywhere. Future research with a larger number of participants could attempt to sort out children as a function of whether their behavior is dominated by geometric features or by the CNN model, and then examine how their brain activity profiles differ.

## Artificial neural networks: limits and promising directions

While simple CNNs are predictive of early ventral visual activity, the present work adds to a growing list of their limits as full models of visual perception (*Bowers et al., 2022*; *Jacob et al., 2021*). Simple CNNs have been shown to fail to model many visual illusions (*Bowers et al., 2022*; *Jacob et al., 2021*), geometrically impossible objects (*Heinke et al., 2021*), global shape, part-whole relations, and other Gestalt properties (*Bowers et al., 2022*). While larger CNNs may overcome some of these issues, and indeed here we found that the ConvNeXT CNN-like architecture could mimic our MEG data, their huge training sets likely include human geometric shapes and drawings, without which they may fail in out-of-domain generalization (*Mayilvahanan et al., 2025*). Human children, by contrast, recognize and produce geometric drawings early on and with little or no explicit training (*Goodenough, 1928*; *Long et al., 2024*; *Saito et al., 2014*). Thus, these observations suggest that, to capture the human sense of geometry, current artificial-intelligence models may have to be supplemented, not only by increasing their size (*Conwell et al., 2021*) and training sets, but also by changing their architecture in ways that better incorporate findings from psychology and neuroscience (*Thompson et al., 2023*; *Biederman, 1987*). Indeed, connectionist models based on the transformer architecture seem to be superior in capturing human geometric perception (*Campbell et al., 2024*), and indeed we found that our results could also be captured by the late layers of visual transformer DINOv2 (*Oquab et al., 2023*), see *Figure 3—figure supplement 1*. This finding may offer an exciting avenue to analyze a mechanistic implementation of symbol-like representations in a connectionist architecture. However, it is also possible that much more sophisticated mechanisms of Bayesian program inference are required to truly capture the remarkable efficiency with which humans grasp abstract shapes and geometric drawings (*Lake et al., 2015*; *Lake et al., 2017*; *Sablé-Meyer et al., 2022*).

## Shape skeleton and principal axis

Skeletal representations have been proposed as human-like representations of visual shapes, particularly appropriate for biological shapes such as animals or plants (*Blum, 1973*). Even for simple geometric shapes, skeletal representation may be automatically and unconsciously computed (*Firestone and*

*Scholl, 2014*). However, in the present work, the two skeletal shape representations we tested did not model human data well (see *Figure 3—figure supplement 1*). Within our quadrilaterals, which are visually similar to one another, all but the square possess the same skeleton topological graph, with only variations in the length and angles of the skeletal segments. Thus, a skeletal representation is probably not the source of the large geometric regularity effect that we observed here and in past work (*Sablé-Meyer et al., 2021*). Still, as previously argued (*Sablé-Meyer et al., 2022*), geometric and medial axis theories need not be seen as incompatible. Rather, we conceive of them as complementary codes, best suited for distinct shape domains. Even after extraction of a figure's skeleton, further compression could be achieved by encoding its geometric regularity, and this may be what humans do when they draw a snake, for instance, as a geometrically regular zigzag.

## Neurophysiological implementation of geometry

Symbolic models are often criticized for lack of a plausible neurophysiological implementation. It is therefore important to discuss whether and how the postulated symbolic geometric code could be realized in neural circuits. There are several distinct challenges. First, some neurons should encode individual features such as lines and curves, or features formed by their relationships (e.g. parallelism, right angle). Second, these codes should be discrete and categorical, forming sharp boundaries between, say, parallel versus non-parallel lines. Third, they should enter into compositional expressions describing how individual features combine into the whole shape (e.g. 'a shape with four equal sides and four equal right-angles').

The first point has been studied in both humans and monkeys in the context of research on the non-accidental properties (NAPs) of objects. NAPs are qualitative features of object shapes, such as straight versus curved, which remain invariant when an object is rotated. Metric properties (MPs), on the other hand, are quantitative properties such as amount of curvature that do not exhibit such invariance. Behavioral research has demonstrated that, for equivalent amounts of pixel change in the image, changes in NAPs are more discriminable than changes in MPs, in human adults with and without formal western education, toddlers, and even in infants (*Amir et al., 2012*; *Biederman et al., 2009*; *Kayaert and Wagemans, 2010*). Furthermore, the firing of neurons in monkey infero-temporal cortex is more sensitive to changes in NAPs than in MPs, whether they are conveyed by 3D shapes (*Kayaert et al., 2003*) or 2D shapes (*Kayaert et al., 2005a*; *Kayaert et al., 2005b*). These findings occurred even in the absence of a task other than passive fixation and without particular training for the stimuli. Further work with 2D shapes, including triangles and quadrilaterals partially overlapping with the present stimuli, showed that IT cortex neurons could also be sensitive to more global shape properties such as axis curvature or 'taper' (the difference between a rectangle seen upfront or at a slanted axis) (*Kayaert et al., 2005a*). Multidimensional scaling of macaque neural population responses organized the tested shapes in a systematic 'shape space' where, for instance, the rectangle occupies an extreme corner, thus making it distinct from its curved or tapered variants (*Kayaert et al., 2005a*).

While these previous non-human primate findings may provide a neurophysiological basis for the fMRI and MEG responses observed here, the neural implementation of our third requirement of having neural codes enter compositional expressions remains elusive. What is more, there are still notable differences between humans and macaques: humans do not need to be extensively trained to understand the differences between geometric shapes (*Izard et al., 2022*; *Izard and Spelke, 2009*), and spontaneously impose geometric categories such as 'parallel' or 'right angle' to a continuum of angles between lines (*Dillon et al., 2019*). Several behavioral findings suggest that a distinct neural code for geometry may exist in humans. Non-human primates perform poorly on a broad variety of perceptual and production tasks with geometric shapes: baboons do not exhibit a human-like geometric complexity effect (*Sablé-Meyer et al., 2021*; *Dehaene et al., 2022*) chimpanzees do not transfer learning of visual categories from concrete pictures to geometric line drawings (*Close and Call, 2015*) and chimpanzees behave very differently from children in free-drawing experiments where they have to complete partial line-drawings of faces (*Saito et al., 2014*). Recent work has argued that crows recognize quadrilaterals in a way that is similar to humans (*Schmidbauer et al., 2025*), though see *Sablé-Meyer and Dehaene, 2025* for a discussion about this result. These findings suggest that an understanding of the mechanisms that underlie the human coding of geometric shapes may ultimately shed light on the cognitive and neural singularity of the human brain (*Dehaene et al., 2022*).

In the future, replicating the present experiments with monkey fMRI and electrophysiology is therefore an important goal. The methodology could also be extended to the perception of a broader set of geometric patterns (circles, spirals, crosses, zigzags, plaids, etc.) which recur since prehistory and in the drawings of children and adults of various cultures, thus testing whether they too originate in a minimal and universal 'language of thought' (*Sablé-Meyer et al., 2022*) and whether such a language is unique to the human species (*Dehaene, 2026*; *Dehaene et al., 2022*). Finally, this research should ultimately be extended to the representation of three-dimensional geometric shapes, for which similar symbolic generative models have been proposed (*Biederman, 1987*; *Leyton, 2003*).

## Methods
### Experiment 1
#### Participants
330 participants (142 females, 177 males, 11 others; mean age 51.1 years, SD = 16.8) were recruited via a link provided on a New York Times article (available at here) which reported previous research from the lab and featured a link to our new online experiment. When participants clicked the link, they landed on a page with our usual procedure for online experiments, including informed consent and demographic questions. No personal identity information was collected.

#### Task
On each trial, participants were shown a 3 × 3 square grid of shapes, eight of which were copies of the same shape up to rotation and scaling, and one of which was a different shape. Participants were asked to detect the intruder shape by clicking on it. Auditory feedback was provided in the form of tones of ascending or descending pitch, as well as coloring of the shapes (the intruder was always colored in green indicating what the right answer was, and in case of an erroneous choice, the chosen shape was colored in red indicating a wrong answer).

#### Stimuli
Shapes design followed previous work exactly (*Sablé-Meyer et al., 2021*): 11 quadrilaterals with a varying number of geometric features, matched for average pairwise distance of all vertices and length of the bottom side. Shapes were presented in pure screen white on pure screen black. Each shape was differently scaled and rotated by sampling nine values without replacement in the following scaling factors [0.85, 0.88, 0.92, 0.95, 0.98, 1.02, 1.05, 1.08, 1.12, 1.15] and rotation angles [–25°, –19.4°, –13.8°, –8.3°, –2.7°, 2.7°, 8.3°, 13.8°, 19.4°, 25°]. Note that the rotation angles were centered on 0° but excluded this value so that the sides of the shapes were never strictly vertical or horizontal. Participants performed 110 trials (11 × 10), one for each pair of different reference and intruder shapes. The order of trials was randomized, subject to the constraint that no two identical reference shapes were used on consecutive trials, and that the outlier of a trial was always different from the reference shape of the previous trial. Two examples of trials are shown in *Figure 1B*.

#### Procedure
The experimental procedure started with instructions, followed by a series of questions: device used (mouse or touchscreen), country of origin, gender, age, highest degree obtained. Participants then provided subjective self-evaluation assessments, with answers on a Likert scale from 1 to 10, for the following items: current skills in mathematics; current skills in first language. Finally, participants performed the task. The instructions text was the following: 'The game is very simple: you will see sets of shapes on your screen. Apart from small rotation and scaling differences, they will be identical, except for one intruder. Your task is to respond as fast and accurately as you can about the location of the intruder by clicking on it. The difficulty will vary, but you always have to answer'.

#### Estimation of empirical representational dissimilarity
To estimate the representational dissimilarity across shapes, first we estimated the dissimilarity between two shapes as the average success rate divided by the average response time. Indeed, if two shapes are very dissimilar, we expect participants to make few mistakes (high success rate) and

find the intruder fast (low response time), yielding a high value, and vice versa. Because we did not have predictions using the asymmetry in visual search (e.g. finding a square within rectangles versus a rectangle versus squares), we then averaged over these paired conditions, thereby turning the square dissimilarity matrix into a triangular dissimilarity matrix. Finally, we z-scored these dissimilarity estimates.

As participants performed a single trial per pair of shapes, the estimation of the dissimilarity is noisy at the single participant level (either 0 or 1/RT depending on whether they answered correctly). We kept this estimate for analyses at the single participant level or mixed-effect analysis; however, for analyses that required a single RDM estimate across participants, we pooled the data from participants to estimate the average success rates and response times, hence before estimating the empirical RDMS, rather than estimating one RDM per participant and then averaging.

## Comparison with model RDMs

To obtain theoretical RDMs with which to compare the present behavioral data, as well as subsequent brain-imaging data, we proceeded as follows. For the CNN encoding model, we downloaded from GitHub the weights for several neural networks [CORnet (*Kubilius et al., 2018*), ResNet (*He et al., 2016*), and DenseNet (*Huang et al., 2018*); all high scoring on brain-score (*Schrimpf et al., 2018*)], all pre-trained with ImageNet and not specifically trained for our task. We extracted the activation vectors in each hidden layer associated with each shape with the $6 \times 6 = 36$ different orientations and scaling used in the experiment. For each shape, we separated their 36 exemplars randomly into two groups to have independent estimation of the representation vectors from the network and used the cross-validated Mahalanobis distance between these two splits to estimate the distance between each pair of shapes. This provides us with an RDM that captures how dissimilar shapes are according to their internal representations in a CNN of object recognition. Unless specified otherwise (see appendix), we report correlation with CORnet layer IT following *Sablé-Meyer et al., 2021*.

For the geometric feature model, we estimate a feature vector of geometric features for each shape. This feature vector includes information about (1) right angles (one for each angle, four features); (2) angle equality (one for each pair of angles, six features); (3) side length equality (one for each pair of sides, six features); and (4) side parallelism (one for each pair of sides, six features); all of this was done up to a tolerance level (e.g. an angle slightly off a right angle still counts as a right angle), using the tolerance value of 12.5% which was fitted previously to independent behavioral data (*Sablé-Meyer et al., 2021*). The dissimilarity between each pair of shapes was the difference between the number of features that each shape possesses: because both the square and the rectangle share many features, they are similar. Two very irregular shapes also end up similar as well. Conversely, the square and an irregular shape end up very dissimilar.

## Experiment 2

### Participants

Twenty healthy French adults (9 females; 19–37 years old, mean age = 24.6 years old, SD: 5.2 years old) and 25 French first graders (13 females; all 6 years old) participated in the study. Three children quit the experiment before any task began because they did not like the MRI noise or lying in the confined space for the scanner. One child completed the localizer task but not the other tasks. One child was missing a single run from the intruder task. All participants had normal hearing, normal or corrected-to-normal vision, and no known neurological deficit. All adults and guardians of children provided informed consent, and adult participants were compensated for their participation.

### Task

In three localizer runs, children and adult participants were exposed to eight different image categories, such that single geometric shapes could be compared to matched displays of faces, houses, and tools, and rows of three geometric shapes could be compared to matched displays of numbers, French words, and Chinese characters. To maintain attention, participants were asked to keep fixating on a green central fixation dot (radius = 8pixels, RGB color = 26, 167, 19, always shown on the screen), and to press a button with their right hand whenever a star symbol was presented. The star spanned roughly the same visual angle as the stimuli from the eight categories and appeared randomly once in

one of the two blocks per category (8 target stars total), between the 3rd to the 6th stimuli within that block. As feedback, a 300 ms 650 Hz beep sound was provided after each button press.

### Stimuli

In each miniblock, participants saw a series of six grayscale images, one per second, belonging to one of eight different categories: faces, houses, tools, numbers, French words, Chinese characters, single geometric shapes, and rows of three geometric shapes. Each category comprised 20 exemplars. All faces, 16 houses, and 18 tools had been used in previous localizer experiments (*Zhan et al., 2018*). For face stimuli, front-view neutral faces (20 identities, 10 males) were selected from the Karolinska Directed Emotional Faces database (*Lundqvist et al., 1998*). The stimuli were aligned by the eyes and the iris distances. A circular mask was applied to exclude the hair and clothing below the neck. House and tool stimuli were royalty-free images obtained from the internet. House stimuli were photos of 2- to 3-story residence houses. Tool stimuli were photos of daily hand-held tools: half of the images were horizontally flipped, so that there were 10 images in a position graspable for the left and right hand, respectively. For French word stimuli, three-letter French words were selected which were known to first graders and had high occurrence frequencies (range = 7–2146 occurrences per million, mean = 302, SD = 505, based on Lexique, http://www.lexique.org/). Chinese characters were selected from the school textbook of Chinese first graders. Chinese word frequency (range = 11–1945 occurrences per million, mean = 326, SD = 451; *Cai and Brysbaert, 2010*) was not significantly different from French words used here ($t(38) = 0.2$, $p = 0.87$). Single-digit formula stimuli were three-character simple operations in the form of 'x + y' or 'x − y' with x greater than y, x ranging from 2 to 5, and y from 1 to 4. Single shapes consisted of a single, centered outline of a geometrical shape (diamond, hexagon, rhombus, parallelogram, rectangle, square, trapezoid, isosceles triangle, equilateral triangle, and right triangle), and were matched in luminance, contrast, and visual angle to the faces/houses/tools/words/Chinese characters which also displayed single objects. A row of shapes consisted of three different shapes side by side, whose total width, size, and line width were matched with three-letter French words and three-character single-digit operations. To match the appearance of the monospaced font in previous work (*Vinckier et al., 2007*), the monospaced font Consolas was used for the French words and numbers, with identical font weight 900. The font for Chinese characters was Heiti, which looks similar to Consolas. Random font size (uniform in 35–55 px font size) was repeatedly sampled until text stimuli achieved similar variability as with the other categories.

The stimuli were embedded in a gray circle (RGB color = 157, 157, 157, radius = 155 pixels), on the screen with a black background. Within the gray circle, the mean luminance and contrast of the 8 stimuli categories did not differ significantly (luminance: $F(7,152) = 0.6$, $p = 0.749$; contrast: $F(7,152) = 1.2$, $p = 0.317$), see *Figure 2—figure supplement 1*.

### Procedure

The eight categories were presented in distinct blocks of 6 s each, fully randomized for block presentation order, with the restriction that there were no consecutive blocks from the same category. Each miniblock comprised six stimuli, in random order. Each stimulus was presented for 1 s, with no interval in between (*Dehaene-Lambertz et al., 2018*). The inter-block interval duration was jittered (4, 6, or 8 s; mean = 6 s). Each of the eight-block types appeared twice within each run. A 6-s fixation period was included at both the beginning and end of the run. Each run lasted for 3 min 24 s, and participants performed three such runs during the fMRI session.

## Experiment 3

### Participants

Identical to experiment 2.

### Procedure and stimuli

#### Task

We adapted the geometric intruder detection task (*Sablé-Meyer et al., 2021*) to the fMRI scanner. On each trial, participants (children and adults) saw an array of six shapes around fixation (three on the right, and three on the left; see *Figure 3A*). Five shapes were identical except for a small amount of

random rotation and scaling, while one was a deviant shape. Because the pointing task used in *Sablé-Meyer et al., 2021* was not possible in the limited space of the fMRI scanner, participants were merely asked to click a button with their left or right hand, thereby indicating on which side they thought the deviant was. Participants responded on every trial, the side of the correct response was counterbalanced within each shape, and we verified that the average motor response side was unconfounded with geometric shape or complexity. After each answer, auditory feedback was provided with a tone of high, increasing pitch when the answer was correct, and a low-pitch tone otherwise.

### Stimuli

Geometric shapes were generated following the procedure described in previous work (*Sablé-Meyer et al., 2021*): to fit the experiment within the time constraints of children fMRI, a subset of shapes was used, comprising square, rectangle, isosceles trapezoid, rhombus, right hinge, and irregular shapes to span the range of complexity found in *Sablé-Meyer et al., 2021*. Following previous work (*Sablé-Meyer et al., 2021*), deviants were generated by displacing the bottom right corner by a constant distance in four possible positions (see *Figure 3A*). That distance was a fraction of the average distance between all pairs of points, which was standardized across shapes (45% change). On each trial, six gray-on-black shapes were shown (shape color rgb values: 127, 127, 127). Shapes were displayed along two semicircles: the positions were determined by positioning the three leftmost (resp. rightmost) shapes on the left side (resp. right side) of a circle of radius 120 px, at angles 0, pi/2, and pi, and then shifting them 100 px to the left (resp. right). The rotation and scaling of each shape were randomized so that no two shapes had the same scaling or rotation factor, and values were sampled in [0.875, 0.925, 0.975, 1.025, 1.075, 1.125] for scaling and [–25°, –15°, –5°, 5°, 15°, 25°] for rotations, avoiding 0° to prevent alignment of specific shapes with screen borders. One of the shapes was an outlier, whose position was sampled uniformly in all six possible positions such that no two consecutive trials featured outliers in the same position. Outliers were sampled uniformly from the four possible types of outliers, so that all outlier types occurred as often, but no two consecutive trials featured identical outlier types.

### Procedure

The six shapes were presented in miniblocks, in randomized order, with no two consecutive blocks with the same type of shape. Each block comprised five consecutive trials with an identical base shape, each with 2 s of stimulus presentation and 2 s of fixation. There was a 4-, 6-, or 8-s delay between blocks. A central green fixation cross was always on display, and it turned bold 600 ms before a block would start. Each run of the outlier detection task lasted 3m40s.

## fMRI methods common to experiments 2 and 3

### MRI acquisition parameters

MRI acquisition was performed on a 3T scanner (Siemens, Tim Trio), equipped with a 64-channel head coil. Exactly 113 functional scans covering the whole brain were acquired for each localizer run, as well as on 179 functional scans covering the whole brain for each run of the geometry task. All functional scans were using a T2*-weighted gradient echo-planar imaging sequence (69 interleaved slices, TR = 1.81 s, TE = 30.4 ms, voxel size = 2 × 2 × 2 mm, multiband factor = 3, flip angle = 71°, phase encoding direction: posterior to anterior). To reconstruct accurate anatomical details, a 3D T1-weighted structural image was also acquired (TR = 2.30 s, TE = 2.98 ms, voxel size = 1 × 1 × 1 mm, flip angle = 9°). To estimate distortions, two spin-echo field maps with opposite phase encoding directions were acquired: one volume in the anterior-to-posterior direction (AP) and one volume in the other direction (PA). Each fMRI session lasted for around 50 min for children including (in order) three runs of a task not discussed here, three Category localizer runs, T1 collection, and two Geometry runs. For adults, the session lasted for around 1 h 20 min because they took the same runs as for children as well as an additional harder version of the geometry task. This version, which involved smaller deviant distances and a stimulus presentation duration of only 200 ms, turned out to be too difficult. While the overall performance still shows a correlation with complexity ($r^2$ = 0.63, p < 0.02), it was entirely driven by two shapes, the square and the rectangle: other shapes were equally hard, and although they were better than chance, they did not correlate with complexity ($r^2$ = 0.35, p = 0.22) while the correlations remained for the simpler condition.

## Data analysis

Preprocessing was performed with the standard pipeline fMRIPrep. Results included in this manuscript come from preprocessing performed using fMRIPrep 20.0.5 (*Esteban et al., 2019*), which is based on Nipype 1.4.2 (*Gorgolewski et al., 2011*; *Gorgolewski et al., 2018*), and generated the following detailed method description.

## Anatomical data

The T1-weighted (T1w) image was corrected for intensity non-uniformity (INU) with N4BiasField-Correction (*Tustison et al., 2010*), distributed with ANTs 2.2.0 (*Avants et al., 2008*), and used as T1w-reference throughout the workflow. The T1w reference was then skull-stripped with a Nipype implementation of the antsBrainExtraction.sh workflow (from ANTs), using OASIS30ANTs as target template. Brain tissue segmentation of cerebrospinal fluid (CSF), white matter (WM), and gray matter (GM) was performed on the brain-extracted T1w using fast (*Zhang et al., 2001*). Brain surfaces were reconstructed using recon-all (*Dale et al., 1999*), and the brain mask estimated previously was refined with a custom variation of the method to reconcile ANTs- and FreeSurfer-derived segmentations of the cortical GM of Mindboggle (*Klein et al., 2017*). Volume-based spatial normalization to two standard spaces (MNI152NLin6Asym, MNI152Nlin2009cAsym) was performed through nonlinear registration with antsRegistration (ANTs 2.2.0), using brain-extracted versions of both T1w reference and the T1w template. The following templates were selected for spatial normalization: FSL's MNI ICBM 152 non-linear 6th Generation Asymmetric Average Brain Stereotaxic Registration Model (*Evans et al., 2012*) and ICBM 152 Nonlinear Asymmetrical template version 2009c (*Fonov et al., 2009*).

## Functional data

For each of the 10 BOLD EPI runs found per subject (across all tasks and sessions), the following preprocessing was performed. First, a reference volume and its skull-stripped version were generated using a custom methodology of fMRIPrep. Susceptibility distortion correction was omitted. The BOLD reference was then co-registered to the T1w reference using bbregister (FreeSurfer) which implements boundary-based registration (*Greve and Fischl, 2009*). Co-registration was configured with six degrees of freedom. Head-motion parameters with respect to the BOLD reference (transformation matrices, and six corresponding rotation and translation parameters) are estimated before any spatio-temporal filtering using mcflirt (*Jenkinson et al., 2002*). BOLD runs were slice-time corrected using 3dTshift from AFNI 20160207 (*Cox and Hyde, 1997*). The BOLD time series (including slice-timing correction when applied) were resampled onto their original, native space by applying the transforms to correct for head motion. These resampled BOLD time series will be referred to as preprocessed BOLD in original space, or just preprocessed BOLD. The BOLD time series were resampled into several standard spaces, correspondingly generating the following spatially normalized, preprocessed BOLD runs: MNI152Nlin6Asym and MNI152Nlin2009cAsym. First, a reference volume and its skull-stripped version were generated using a custom methodology of fMRIPrep. Several confounding time series were calculated based on the preprocessed BOLD: framewise displacement (FD), DVARS, and three region-wise global signals. FD and DVARS are calculated for each functional run, both using their implementations in Nipype (following the definitions by *Power et al., 2014*). The three global signals are extracted within the CSF, the WM, and the whole-brain masks. Additionally, a set of physiological regressors was extracted to allow for component-based noise correction (*Behzadi et al., 2007*). Principal components are estimated after high-pass filtering the preprocessed BOLD time-series (using a discrete cosine filter with 128 s cut-off) for the two CompCor variants: temporal (tCompCor) and anatomical (aCompCor). tCompCor components are then calculated from the top 5% variable voxels within a mask covering the subcortical regions. This subcortical mask was obtained by heavily eroding the brain mask, which ensures it does not include cortical GM regions. For aCompCor, components are calculated within the intersection of the aforementioned mask and the union of CSF and WM masks calculated in T1w space, after their projection to the native space of each functional run (using the inverse BOLD-to-T1w transformation). Components are also calculated separately within the WM and CSF masks. For each CompCor decomposition, the k components with the largest singular values are retained, such that the retained components' time series are sufficient to explain 50% of variance across the nuisance mask (CSF, WM, combined, or temporal). The remaining components are dropped from consideration. The head-motion estimates calculated in the correction step were also

placed within the corresponding confounds file. The confound time series derived from head motion estimates and global signals were expanded with the inclusion of temporal derivatives and quadratic terms for each (*Satterthwaite et al., 2013*). Frames that exceeded a threshold of 0.5 mm FD or 1.5 standardized DVARS were annotated as motion outliers. All resamplings can be performed with a single interpolation step by composing all the pertinent transformations (i.e. head-motion transform matrices, susceptibility distortion correction when available, and co-registrations to anatomical and output spaces). Gridded (volumetric) resamplings were performed using antsApplyTransforms (ANTs), configured with Lanczos interpolation to minimize the smoothing effects of other kernels (*Lanczos, 1964*). Non-gridded (surface) resamplings were performed using mri_vol2surf (FreeSurfer).

Many internal operations of fMRIPrep use Nilearn 0.6.2 (*Abraham et al., 2014*), mostly within the functional processing workflow.

## fMRI GLM models

fMRI first-level models (GLM) were computed by convolving the experimental design matrix (specific to each task, see below) with SPM's HRF model as implemented in Nilearn, with the following parameters: spatial smoothing using a full width at half maximum window (fwhm) of 4 mm; a second-order autoregressive noise model; and signal standardized to percent signal change relative to whole-brain mean. The following confound regressors were added: polynomial drift models from constant to fifth order (six regressors); estimated head translation and rotation on three axes (six regressors) as well as the following confound regressors given by fmriprep: average cerebro-spinal fluid signal (one regressor), average WM signal (one regressor), and the first five high-variance confounds estimates (*Behzadi et al., 2007*) (five regressors).

In the visual category localizer, the design matrix contained distinct events for each visual stimulus, grouped by category, each with a duration of 1 s, including a specific one for the target star, leading to nine regressors (Chinese, face, house, tools, numbers, words, single shape, three shapes, and star). In the geometry task runs, each reference shape was associated with a regressor with trial duration set to 2 s, thus leading to 6 regressors (square, rectangle, rhombus, iso-trapezoid, hinge, and random). In both cases, button presses were not modeled as they were fully correlated with predictors of interest (either the star for the localizer, or every single trial for the geometry task).

Second-level models were estimated after additional smoothing with a full width at half maximum window of 8 mm. Statistical significance of clusters was estimated with a bootstrap procedure as follows: given an uncorrected p-value of 0.001, clusters were identified by contiguity of voxels that had p-values below this threshold. Then, the p-value of a cluster was derived by comparing its statistical mass (the sum of its t-values) to the distribution of the maximum statistical mass obtained by performing the same contrast after randomly swapping the sign of each participant's statistical map. We performed this swapping 10,000 times for each contrast we estimated and computed the corrected p-value accordingly; for instance, a cluster whose mass was only outperformed by 3 random swaps out of the 10,000 was assigned a p-value of 0.0003.

## Searchlight RSA analyses

First, we estimated the RDM within spheres centered on each voxel. For this, we performed a searchlight sweep across the whole brain. We extracted the GLM coefficients of the geometric shapes from each voxel and all the neighboring voxels in a 3-voxel radius (=6 mm), for a total of 93 voxels per sphere. We discarded voxels where more than 50% of this sphere fell outside the participant's brain. We extracted the betas of each shape and used a cross-validated Mahalanobis distance across runs (crossnobis, implemented in rsatoolbox) to estimate the dissimilarity between each pair of shapes. We attributed this distance to the center of the searchlight, thereby estimating an empirical RDM at each location.

Then we compared this empirical RDM with the two RDMs derived from our two models separately, using a whitened correlation metric. Both choices of metrics (cross-nobis and whitened correlation) follow the recommendations from a previous methodological publication (*Diedrichsen et al., 2021*).

Finally, we computed group-level statistics by smoothing the resulting correlation map (fwhm = 8 mm) and performing statistical maps, cluster identification, and statistical inference at the cluster level as we did for the second-order level analysis, but with a one-tailed comparison only as we did not consider negative correlations.

The bar plot for ROIs in *Figure 2—figure supplement 3* reflects a subject-specific voxel localization within ROIs. Within each ROI identified, we find, for each subject, the 10% most responsive subject-specific voxels in the same contrast used to identify the cluster. To avoid double-dipping, we selected these voxels using the contrast from one run, then collected the fMRI responses (beta coefficients) from the other runs; we perform this procedure across all runs and average the responses. Error bars indicate the standard error of the mean across participants.

## Experiment 4

### Participants

Twenty healthy French adults (13 females; 21–42 years old, mean: 24.9 years old, SD: 8.1 years old) participated in the MEG study. All participants had normal hearing, normal or corrected-to-normal vision, and no neurological deficit. All adults provided informed consent and were compensated for their participation. For all but one participant, we had access to anatomical recordings in 3T MRI, either from prior, unrelated experiments in the lab, or because the MEG session was immediately followed by a recording. Because of one participant missing an anatomical recording, analyses that required source reconstruction were performed on 19 subjects.

### Task

During MEG, adult participants were merely exposed to shapes while maintaining fixation and attention. As in previous work (e.g. *Al Roumi et al., 2023*; *Benjamin et al., 2024*), the goal was to examine the spontaneous encoding of stimuli and the presence or absence of a novelty response to occasional deviants.

### Stimuli

All 11 geometric shapes in *Figure 1A* were presented in miniblocks of 30 shapes. Geometric shapes were presented centered on the screen, one shape every second, with shapes remaining onscreen for 800 ms and a centered fixation cross present between shapes for 200 ms. To make the shapes more attractive (and since the same shape was also used in an infant experiment, not reported here), during their 800 ms presentation, the shapes slowly increased in size: in total, a scale factor of 1.2 was applied over the course of 800 ms, with linear interpolation of the shape size during the duration of the presentation. Shapes were presented in miniblocks following an oddball paradigm: within a miniblock, all shapes were identical up to scaling (randomly sampled in [0.875, 0.925, 0.975, 1.025, 1.075, 1.125]) and rotation (sampled in [–25°, –15°, –5°, 5°, 15°, 25°]), except for occasional oddballs which were deviant versions of the reference shape. Each miniblock comprised 30 shapes, 4 of which were oddballs that could replace any shape after the first six. Two oddballs never appeared in a row. There was no specific interval between miniblocks beyond the usual duration between shapes. A run was made of 11 miniblocks, one per shape in random order, and participants attended 8 runs except two participants who are missing a single run due to experimenters' mistakes when setting up the MEG acquisition.

### Procedure

After inclusion by the lab's recruiting team, participants were prepared for the MEG with electrocardiogram (ECG) and electrooculogram (EOG) sensors, as well as four Head Position Indicator coils, which were digitalized to track the head position throughout the experiment. Then we explained to participants that the task was a replication of an experiment with infants, and therefore was purely passive: they would be shown shapes and were instructed to pay attention to each shape, while trying to avoid blinks as well as body and eye movements. They sat in the MEG and we checked the head position, ECG/EOG, and MEG signal. From that point onward, we never opened the MEG door again to avoid resetting MEG signals and allow for optimal cross-run decoding and generalization. Participants took typically eight runs consecutively, with small breaks with no stimuli between runs to rest their eyes. At the end of the experiment, participants took the intruder test from previous work (*Sablé-Meyer et al., 2021*) on a laptop computer outside of the MEG, and finally, we spent some time debriefing with participants the goal of the experiment.

To ensure high accuracy of the timing in the MEG, each trial's first frame contained a white square on the bottom of the screen, which was hidden from participants but recorded with a photodiode. The same area was black during the rest of the experiment. The ramping up of the photodiode was therefore synchronized with the screen update and the appearance of the stimulus, ensuring robust timing for analyses. Then each 'screen update' event was linked to a recorded list of presented shapes.

## MEG acquisition parameters

Participants were instructed to look at a screen while sitting inside an electromagnetically shielded room. The magnetic component of their brain activity was recorded with a 306-channel, whole-head MEG by Elekta Neuromag (Helsinki, Finland). The MEG helmet is composed of 102 triplets of sensors, each comprising one magnetometer and two orthogonal planar gradiometers. The brain signals were acquired at a sampling rate of 1000 Hz with a hardware high-pass filter at 0.03 Hz.

Eye movements and heartbeats were monitored with vertical and horizontal EOGs and ECGs. Head shape was digitized using various points on the scalp as well as the nasion, left and right preauricular points (FASTTRACK, Polhemus). Subjects' head position inside the helmet was measured at the beginning of each run with an isotrack Polhemus Inc system from the location of four coils placed over frontal and mastoid skull areas.

## Preprocessing of MEG signals

The preprocessing of the data was performed using MNE-BIDS-Pipeline, a streamlined implementation of the core ideas presented in the literature (*Jas et al., 2018*) and leveraging BIDS specifications (*Niso et al., 2018*; *Pernet et al., 2019*). The pipeline performed automatic bad channel detection (both noisy and flat), then applied Maxwell filtering and Signal Space Separation on the raw data (*Taulu and Kajola, 2005*). The data was then filtered between 0.1 and 40 Hz and resampled to 250 Hz. Extraction of epochs was performed for each shape, starting 150 ms before stimulus onset and stopping 1150 ms after, and the relevant metadata (event type, run, trial index, etc.) for each epoch was recovered from the stimulation procedure at this step. Artifacts in the data (e.g. blinks and heartbeats) were repaired with signal-space projection (*Uusitalo and Ilmoniemi, 1997*), and thresholds derived with 'autoreject global' (*Jas et al., 2017*). For source reconstruction, some preprocessing steps were performed by fmriprep (see below). Then, sources were positioned using the 'oct5' spacing with 1026 sources per hemisphere, and we used the e(xact)LORETA method (following recommendations from the literature *Jatoi et al., 2014*; *Pascual-Marqui et al., 2018*) using empty-room recordings performed right before or right after the experiment to estimate the noise covariance matrix. Additionally, for source reconstruction, anatomical MRI was preprocessed with fmriprep. T1-weighted (T1w) images were corrected for INU with N4BiasFieldCorrection (*Tustison et al., 2010*), distributed with ANTs 2.3.3 (*Avants et al., 2008*). The T1w reference was then skull-stripped with a Nipype implementation of the antsBrainExtraction.sh workflow. Brain tissue segmentation of CSF, WM, and GM was performed on the brain-extracted T1w using fast (*Zhang et al., 2001*). Brain surfaces were reconstructed using recon-all (*Dale et al., 1999*) and the brain mask estimated previously was refined with a custom variation of the method to reconcile ANTs- and FreeSurfer-derived segmentations of the cortical GM of Mindboggle (*Klein et al., 2017*).

## Decoding

After epoching the data, for each timepoint within an epoch and each participant, we trained a logistic regression decoder to classify epochs as reference or oddball using samples from all shapes. For this analysis, we discarded the six first trials at the beginning of each block, since (1) those could not be oddballs ever and (2) there was no warning of transitions between blocks and so the first trials were also 'oddballs' with respect to the previous block's shape. Each epoch was normalized before training the classifier. To avoid decoders being biased by the overall signal's autocorrelation across timescales, we used sixfolds of cross-validation over runs with the following folds: even runs versus odd runs; first half versus second half; and runs 1, 2, 5, 6 versus 3, 4, 7, and 8. Folds were used in both directions, for example even for training and odd for testing; as well as odd for training and even for testing. The decoders were trained on data from all of the shapes conjointly. When testing its accuracy, we tested it separately on data from each shape (e.g. detecting oddballs within squares only, within rectangles only, etc.) – using runs independent from the training data. In order to estimate

accuracy without being biased by the imbalanced number of epochs in the different classes (there are 4 oddballs for every 20 references), we report the ROC area under the curve of the receiver operating characteristic for each shape in *Figure 4B*, left subfigure. Then at each timepoint, we correlated the decoding performance with the number of geometric features present in each shape: the r correlation coefficient at each timepoint is plotted in the central subfigure, together with shading for a significant cluster identified with permutation tests across participants (implemented by mne, one-tailed, $2^{13}$ permutations, cluster forming threshold <0.05). Finally, we average the decoding performance in the identified cluster and plot each shape's average decoding performance against online behavioral data: this is effectively visualizing the same data as the central column's figure, and therefore no statistical test is reported. The analyses were performed both without any smoothing and with a uniform averaging over a sliding window of 100 ms; the results are identical, but we chose the latter since plots using the smoothed version make the separation of the different shapes easier to see. The same holds true for the next analysis.

In *Figure 4C*, we display a similar sequence of plots, but now instead of training a single classifier to identify epochs as reference or oddball conjointly on all shapes, we train 11 separate such classifiers, one for each shape. This produces very similar results from the previous analysis.

## RSA analysis

For RSA analyses, data from the oddballs was discarded and we used data from the magnetometers only. The goal of this analysis was to pinpoint when and where the mental representation of shapes followed distances that matched either a neural network model or a geometric feature model. We used the same model RDMs as the one we used to analyze behavior and fMRI data and provide below the details of how we derived the empirical RDMs for our analyses.

### In sensor space

We estimated, at each timepoint, the dissimilarity between each pair of shape across sensors: we relied on rsatoolbox to compute the Mahalanobis distance, cross-validated with a leave-one-out scheme over runs. This provided us with one empirical RDM for each timepoint, with no spatial information as this was performed across all sensors. Then we compared this RDM with our two models: since our model RDMs are effectively orthogonal, we performed the comparisons with the empirical RDM separately. We used a whitened correlation metric to compare RDMs. This gave us one timeseries for each participant and each model, and we then performed permutation testing in the [0, 800]ms window to identify significant temporal clusters for each model separately. As shown in *Figure 4C*, this yields one significant cluster associated with the CNN encoding model and two significant clusters associated with the geometric feature model.

### In source space

In order to understand not only the temporal dynamic of the mental representations, but also get an estimate of the localization of the various representations, we turned to RSA analysis in source space. We performed source reconstruction of each reference shape trial. Then we averaged the data over the two identified temporal clusters from the sensor-space RSA analysis (we merged the two clusters associated with the exact geometric feature model): [60, 320] and [128, 400] ms. We then performed RSA analysis at each source's location using its neighboring sources (geodesic distance of 2 cm on the reconstructed cortical surface). Finally, we compared the resulting RDMs to either the CNN encoding model or the geometric feature model. These steps were performed independently for each subject. Next, we projected the whitened correlation distance between empirical RDMs and model RDM onto a common cortical space, fsaverage. Finally, we performed a permutation spatial cluster test across participants, using adjacency matrices between sources (sources are adjacent if they were immediate neighbors on the reconstructed surface's mesh). This resulted in two significant clusters associated with the CNN encoding model during the [60, 320] ms time window, located in bilateral occipital areas. Additionally, this resulted in two significant clusters associated with the geometric feature model during the [128, 400] ms, located in very broad bilateral networks encompassing dorsal and frontal areas.

## Acknowledgements

We are grateful to Ghislaine Dehaene-Lambertz, Leila Azizi, and the NeuroSpin support teams for help in data acquisition, and Lorenzo Ciccione, Christophe Pallier, Minye Zhan, Alexandre Gramfort, and the MNE team for support in data processing.

## Additional information

### Funding

| Funder | Grant reference number | Author |
|---|---|---|
| Fondation Fyssen | | Mathias Sablé-Meyer |
| European Research Council | MathBrain | Stanislas Dehaene |
| Institut National de la Santé et de la Recherche Médicale | | Stanislas Dehaene |
| Commissariat à l'Énergie Atomique et aux Énergies Alternatives | | Stanislas Dehaene |
| Collège de France | | Stanislas Dehaene |

The funders had no role in study design, data collection, and interpretation, or the decision to submit the work for publication.

### Author contributions

Mathias Sablé-Meyer, Conceptualization, Data curation, Software, Formal analysis, Validation, Investigation, Visualization, Methodology, Writing – original draft, Writing – review and editing; Lucas Benjamin, Data curation, Formal analysis, Visualization, Writing – review and editing; Cassandra Potier Watkins, Data curation, Project administration, Writing – review and editing; Chenxi He, Data curation; Maxence Pajot, Théo Morfoisse, Formal analysis, Writing – review and editing; Fosca Al Roumi, Supervision, Methodology, Project administration, Writing – review and editing; Stanislas Dehaene, Conceptualization, Resources, Supervision, Funding acquisition, Validation, Investigation, Visualization, Writing – original draft, Project administration, Writing – review and editing

### Author ORCIDs

Mathias Sablé-Meyer ⓘ https://orcid.org/0000-0003-0844-0775
Lucas Benjamin ⓘ https://orcid.org/0000-0002-9578-6039
Fosca Al Roumi ⓘ https://orcid.org/0000-0001-9590-080X

### Ethics

All studies were conducted in accordance with the Declaration of Helsinki and French bioethics laws. On-line collection of behavioral data was approved by the Paris-Saclay University Committee for Ethical Research (reference: CER CER-Paris-Saclay-2019-063). Behavioral and brain-imaging studies in the lab in adults and 6-year-old children (MEG and fMRI) were approved by the CEA ethics boards and by a nationally approved ethics committee (reference MEG: CPP 100049; fMRI in children: CPP 100027 and CPP 100053; fMRI in adults: CPP 100050).

Reviewer #1 (Public review): https://doi.org/10.7554/eLife.106464.3.sa1
Reviewer #2 (Public review): https://doi.org/10.7554/eLife.106464.3.sa2
Reviewer #3 (Public review): https://doi.org/10.7554/eLife.106464.3.sa3
Author response https://doi.org/10.7554/eLife.106464.3.sa4

### Data availability

Scripts for all of the analyses are available at https://github.com/mathias-sm/AGeometricShapeRegularityEffectHumanBrain (copy archived at *Sablé-Meyer, 2025*); behavioral data and scripts to generate

the models are also available at this url. Raw fMRI data is provided at https://openneuro.org/datasets/ds006010/versions/1.0.1 and raw MEG data at https://openneuro.org/datasets/ds006012/versions/1.0.1.

The following datasets were generated:

| Author(s) | Year | Dataset title | Dataset URL | Database and Identifier |
|---|---|---|---|---|
| Sablé-Meyer M, Benjamin L, Watkins CP, He C, Pajot M, Morfoisse T, Roumi FA, Dehaene S | 2025 | A geometric shape regularity effect in the human brain: fMRI dataset | https://openneuro.org/datasets/ds006010/versions/1.0.1 | OpenNeuro, 10.18112/openneuro.ds006010.v1.0.1 |
| Sablé-Meyer M, Benjamin L, Watkins CP, He C, Pajot M, Morfoisse T, Roumi FA, Dehaene S | 2025 | A geometric shape regularity effect in the human brain: MEG dataset | https://openneuro.org/datasets/ds006012/versions/1.0.1 | OpenNeuro, 10.18112/openneuro.ds006012.v1.0.1 |

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

# Appendix

## Additional results in the behavior task: comparison with other CNNs

In *Figure 1—figure supplement 1*, we report additional results from correlating human behavior with various models (top; DenseNet and ResNet) and various layers of a single model (bottom; many layers of CORnet). In all cases, the CNN encoding model is a much worse predictor of the human behavior than the geometric feature model (all p < 0.001). The late layers of CORnet, DenseNet, and ResNet all capture some of the variance of participants' behaviors to comparable; while ResNet is the highest scoring in this analysis, we can see in *Figure 1—figure supplement 1A* that they are all highly correlated, and in order to stay close to previous work, we used CORnet's layer IT throughout the article.

Additionally, the fit of the successive layers of CORnet (V1, V2, V4: small effect sizes, only significant for V1; IT, flatten: much higher effect sizes, increasing between IT and flatten) indicates that the feed-forward processing of the visual input by the CNN encoding yields internal representations that are increasingly closer to humans'. This suggests that even the visual model goes beyond local properties that V1 would capture – but in all cases, the fit is much worse than the geometric feature model.

## Intruder task during fMRI

Overall, both age groups performed better than chance (all p < 0.001) at detecting the intruder, with an average response time of 1.03 s in adults, 1.36 s in children, for the easy condition, and 0.79 s in adults in the hard condition. Both error rates and response times are modulated by the reference shape (all p < 0.001), and both are correlated with error rate data from outside the scanner (all p < 0.05).

## Intruder task in MEG participants

Participants recruited for the MEG study performed no behavioral task inside the MEG, as we relied on a passive presentation oddball paradigm. After the scanner session, participants took one run of the intruder detection task previously used online and described in *Sablé-Meyer et al., 2021* – we presented them with the task after the scanning session to avoid biasing participants toward actively looking for intruders in the oddball paradigm. At the group level, the data fully replicated the geometric regularity effect (*Sablé-Meyer et al., 2021*) (correlation of error rate with that of previous participants, regression over 11 shapes: $r^2$ = 0.90, p < 0.001). A mixed-effect model correlating the error rates of the two groups, with both the slope and the intercept as separate random effects, yielded an intercept not significantly different from 0 (p = 0.42), and a slope significantly different from 0 (p < 1e−10) and not significantly different from 1 (*P*=.19) suggesting that the data was similar in the two groups. We also correlated each participant's average error rate per shape to the group data from our previous dataset (*Sablé-Meyer et al., 2021*). A one-tailed test for a positive slope indicated that 19 out of 20 participants displayed a significant geometric regularity effect. We still included the data from the participant that did not display a significant effect, as we had not decided on such a rejection criterion beforehand.

## fMRI contrasts for geometric shape in the visual localizer

In order to isolate the brain responses to geometric shapes, we focused on the simplest possible contrast, that is greater activation to the presentation of a single geometric shape versus all of its single-image controls (face, house, and tool). This contrast is presented in *Figure 2C*. Note that we excluded Chinese characters from this comparison because they often include geometric features (e.g. parallel lines), but including them gave virtually identical results (compare *Figure 2—figure supplement 2*, identical list of cluster-level corrected clusters at the p < 0.05 level in both age groups). We also included in the design rows of three distinct geometric shapes (e.g. square, triangle, and circle). Our logic here was that this condition, although somewhat artificial from the geometric viewpoint, could be very tightly matched with two other conditions, namely a string of three letters ('words' condition, e.g. BON) or a small three-symbol mathematical operation ('numbers' condition, e.g. 3 + 1). However, the corresponding contrast (three geometric shapes > words and numbers did not give any positive activation: no positive cluster was significant at the whole-brain cluster-level

corrected p < 0.05 level in either age group; additionally, none of the regions of interest (ROIs) identified in the single shape contrast was significant for this contrast: aIPS left, p = 0.975 for adults and p = 0.389 for children; aIPS right, p = 0.09 in both adults and children; pITG right, p = 0.13 in adults and p = 0.362 in children). We reasoned, however, that numbers should be excluded from this contrast because, by our very hypothesis, geometric shapes should activate a number-based program-like representations (e.g. square = repeat(4){line, turn}) (*Sablé-Meyer et al., 2022*). When restricting the contrast to 'three geometric shapes > words', at the whole-brain level, no cluster reached significance at the p < 0.05 level (cluster-level correction). However, in this case, testing the ROIs identified in the single shape condition revealed that while the left aIPS still did not reach significance at the p < 0.05 level (p = 0.71 in adults, p = 0.30 in children), both the right aIPS and the right pITG reached significance (aIPS right: p < 0.001 in adults and p = 0.014 in children; pITG, p = 0.008 in adults and p = 0.046 in children).

## Additional ROI analysis of the ventral pathway in fMRI (*Figure 2—figure supplement 3*)

We used individually defined ROIs to probe the fMRI response to geometric shapes in four cortical regions defined by their preferential responses to four other visual categories known to be selectively processed in the ventral visual pathway: faces, houses, tools, and words. To this aim, we first identified, at the group level, clusters of voxels associated with each visual category. Such clusters were identified using a non-parametric permutation test across participants at the whole-brain level using a contrast for the target category against the mean of the other three (voxel p < 0.001 then clusters p < 0.05 except for fusiform face area [FFA] in children, p = 0.09, and the absence of a well-identified visual word form area [VWFA] in children). Significant clusters that intersect the MNI coordinate z = −14 are shown in *Figure 2—figure supplement 3*; in the case of the FFA and the VWFA, we restricted ourselves to clusters at MNI coordinates typically found in the literature, respectively (−45, −57, −14) and (40, −55, −14).

Then, within each such ROI identified in adults, we identified for each subject, including children, the 10% most responsive subject-specific voxels in the same contrast used to identify the cluster. To avoid double-dipping, we selected the voxels using the contrast from a single run, then collected the fMRI responses (beta coefficients) to all categories from the other runs, and then replicated this procedure across all runs while averaging the responses to a given category. The average coefficients within each such individually defined cortical ROI are shown in *Figure 2—figure supplement 3*, separately for children and adults. Several observations are in order, and detailed below for each visual category.

In the left-hemispheric VWFA, we can see that voxels are indeed responsive to written words in the participants' language (French), more than to an unknown language (Chinese), in both adults and children (paired *t*-tests, p < 0.001 in adults, p = 0.003 in children). VWFA voxels also responded to the symbolic display of numbers entering into small computations (e.g. 3 + 1) in adults, but this response did not appear to be developed yet in children. VWFA voxels also showed a response to tools, particularly in young children, as has already been reported (*Dehaene-Lambertz et al., 2018*). In the opposite direction, they were particularly under-activated by houses in adults. Finally, and most crucially, geometric shapes, whether presented alone or in a string of 3, did not elicit a strong response, indeed no stronger than non-preferred categories such as Chinese characters or faces (p = 0.37 in adults, p = 0.057 in children on a one-tailed paired *t*-test).

In the right-hemispheric FFA, we only saw a purely selective response to faces in both adults and children. All the other visual categories yielded an equally low level of activity in this area. In particular, the responses to geometric shapes were not significantly different from those to other visual categories.

In the bilateral ROIs responsive to tools, a very similar result was found: apart from their strong response to tools and their slightly increased activation to houses and reduced activations to written words and numbers in the right hemisphere, these voxels elicit activations that were essentially equal across all non-preferred visual categories, with geometric shapes being no exception.

Finally, bilateral house-responsive ROIs corresponding to the parahippocampal place area (PPA) were also mildly responsive to tools in both populations and both hemispheres. However,

geometric shapes did not activate these clusters more than other visual categories such as faces (both hemispheres, both age groups have p > 0.05 in one-tailed paired *t*-tests).

## Additional analysis: evoked response potentials

Different participants can, in principle, end up with very different decoder weights, since the decoders are trained independently for each participant. Several of our analyses are based on the decoders' performance only, and therefore the decoder's weights associated with each sensor are not considered. To stay closer to the MEG data, we replicated the previous analysis directly from evoked potential data in the gradiometers. Using spatiotemporal permutation testing, we identified a set of sensors and timepoints across participants where the reference and the oddball epochs were significantly different. We identified three significant clusters: [268, 844], [440, 896], and [492, 896] ms.

Then we computed the average difference between the reference and the oddball trials across these sensors separately for each shape. Finally, we correlated the differences with the number of geometric features in the reference shape. The first cluster, from 268 to 844 ms, did not elicit any significant correlation with the geometric regularity; however, the two others yielded significant clusters at the p < 0.05 level, although with later timing than the decoding analysis, at around ~600 ms.

## MEG: joint MEG-fMRI RSA

Representational similarity analysis also offers a way to directly compare similarity matrices measured in MEG and fMRI, thus allowing for fusion of those two modalities and tentatively assigning a 'time stamp' to distinct MRI clusters (*Cichy and Oliva, 2020*). However, we did not attempt such an analysis here for several reasons. First, distinct tasks and block structures were used in MEG and fMRI. Second, a smaller list of shapes was used in fMRI, as imposed by the slower modality of acquisition. Third, our study was designed as an attempt to sort out between two models of geometric shape recognition. We therefore focused all analyses on this goal, which could not have been achieved by direct MEG–fMRI fusion, but required correlation with independently obtained model predictions.

**Appendix 1—table 1.** Coordinates and characteristics of significant fMRI clusters responding to geometric shapes in localizer runs.

For each age group, each line gives the peak coordinates, volume, and statistics of a cluster with p < 0.05 (whole brain, permutation test) for the contrast 'single shape > other single visual categories'. The sign of the peak *t*-value and the shading indicate whether the contrast was positive (white background) or negative (gray background). Coordinates are given in MNI space.

| Age group | X | Y | Z | Peak *t*-value | Volume in cm² | Cluster corrected p-value |
|---|---|---|---|---|---|---|
| Adults | 51.5 | −54.5 | −8.5 | 5.12 | 4.4 | <0.01 |
| | 35.5 | −48.5 | 55.5 | 4.93 | 7.2 | <0.01 |
| | 39.5 | −76.5 | −12.5 | −6.33 | 51.4 | <0.01 |
| | −34.5 | −70.5 | −10.5 | −5.92 | 42 | <0.01 |
| Children | −12.5 | −76.5 | −48.5 | 5.51 | 3.9 | 0.01 |
| | 55.5 | −24.5 | 45.5 | 5 | 8.3 | <0.01 |
| | 21.5 | −80.5 | −46.5 | 4.9 | 4.4 | 0.01 |
| | −42.5 | −38.5 | 35.5 | 4.9 | 6.4 | <0.01 |
| | 21.5 | −94.5 | −8.5 | −6.34 | 40.1 | <0.01 |
| | −24.5 | −98.5 | −8.5 | −6.17 | 55.3 | <0.01 |

**Appendix 1—table 2.** Coordinates and characteristics of significant fMRI clusters in the RSA analysis.

Coordinates and characteristics of significant fMRI clusters in the RSA analysis. Same organization as *Appendix 1—table 1* for the RSA analysis.

| Age group | Model | X | Y | Z | Peak *t*-value | Volume in cm² | Cluster corrected p-value |
|-----------|-------|---|---|---|----------------|---------------|---------------------------|
| Adults | Geometric features | –0.5 | 13.5 | 49.5 | 5.4 | 64.4 | <0.01 |
| | | –22.5 | –54.5 | 51.5 | 5.38 | 13.6 | <0.01 |
| | | –14.5 | –66.5 | 5.5 | 5.28 | 6.1 | <0.01 |
| | | 31.5 | 31.5 | –8.5 | 4.86 | 2.2 | 0.03 |
| | | –26.5 | –2.5 | 49.5 | 4.79 | 5.3 | <0.01 |
| | | –8.5 | –72.5 | –38.5 | 4.62 | 7 | <0.01 |
| | | –34.5 | 23.5 | 5.5 | 4.15 | 3.2 | <0.01 |
| | | –10.5 | 71.5 | 3.5 | 4.03 | 1.9 | 0.03 |
| | | –46.5 | –0.5 | 33.5 | 3.88 | 1.6 | 0.04 |
| | | 21.5 | –98.5 | –6.5 | 3.72 | 2.3 | 0.02 |
| | CNN encoding | 23.5 | –14.5 | 57.5 | 5.4 | 2.4 | 0.03 |
| | | –48.5 | –82.5 | –0.5 | 4.96 | 4.4 | <0.01 |
| | | –30.5 | –80.5 | 25.5 | 4.55 | 2.9 | 0.02 |
| | | 1.5 | 21.5 | 45.5 | 4.54 | 3.8 | <0.01 |
| | | 27.5 | 33.5 | 5.5 | 4.51 | 3.7 | <0.01 |
| | | 45.5 | –80.5 | –2.5 | 4.38 | 2.2 | 0.03 |
| | | 53.5 | 13.5 | 31.5 | 4.05 | 3.9 | <0.01 |
| Children | CNN encoding | –22.5 | –84.5 | 11.5 | 4.59 | 1.4 | 0.06 |
| | | 43.5 | –82.5 | 11.5 | 4.29 | 2.2 | 0.02 |

