## [Editor Report · eLife Assessment]

This **important** series of studies provides converging results from complementary neuroimaging and behavioral experiments to identify human brain regions involved in representing regular geometric shapes and their core features. Geometric shape concepts are present across diverse human cultures and possibly involved in human capabilities such as numerical cognition and mathematical reasoning. Identifying the brain networks involved in geometric shape representation is of broad interest to researchers studying human visual perception, reasoning, and cognition. The evidence supporting the presence of representation of geometric shape regularity in dorsal parietal and prefrontal cortex is **solid**, but does not directly demonstrate that these circuits overlap with those involved in mathematical reasoning. Furthermore, the links to defining features of geometric objects and with mathematical and symbolic reasoning would benefit from stronger evidence from more fine-tuned experimental tasks varying the stimuli and experience.

---

## [Referee Report · Reviewer #1 (Public review)]

This paper examines how geometric regularities in abstract shapes (e.g., parallelograms, kites) are perceived and processed in the human brain. The manuscript contains multimodal data (behavior, fMRI, MEG) from adults and additional fMRI data from 6-year-old children. The key findings show that (1) processing geometric shapes lead to reduced activity in ventral areas in comparison to complex stimuli and increased activity in intraparietal and inferior temporal regions, (2) the degree of geometric regularity modulates activity in intraparietal and inferior temporal regions, (3) similarity in neural representation of geometric shapes can be captured early by using CNN models and later by models of geometric regularity. In addition to these novel findings, the paper also includes a replication of behavioral data, showing that the perceptual similarity structure amongst the geometric stimuli used can be explained by a combination of visual similarities (as indexed by feedforward CNN model of ventral visual pathway) and geometric features. The paper comes with openly accessible code in a well-documented GitHub repository and the data will be published with the paper on OpenNeuro.

In the revised version of this manuscript, the authors clarified certain aspects of the task design, added critical detail to the description of the methods, and updated the figures to show unsmoothed data and variability across participants. Importantly, the authors thoroughly discussed potential task effects (for the fMRI data only) and added additional analyses that indicate that the effects are unlikely to be driven by linguistic labels/name availability of the stimuli.

Comments on the revision:

Thank you for carefully addressing all my concerns and especially for clarifying the task design.

---

## [Referee Report · Reviewer #2 (Public review)]

Summary

The current study seeks to understand the neural mechanisms underlying geometric reasoning. Using fMRI with both children and adults, the authors found that contrasting simple geometric shapes with naturalistic images (faces, tools, houses) led to responses in the dorsal visual stream, rather than ventral regions that are generally thought to represent shape properties. The author's followed up on this result using computational modeling and MEG to show that geometric properties explain distinct variance in the neural response than what is captured by a CNN.

Strengths

These findings contribute much-needed neural and developmental data to the ongoing debate regarding shape processing in the brain and offer additional insights into why CNNs may have difficulty with shape processing. The motivation and discussion for the study is appropriately measured, and I appreciate the authors' use of multiple populations, neuroimaging modalities, and computational models in explore this question.

Weaknesses

The presence of activation in aIPS led the authors to interpret their results to mean that geometric reasoning draws on the same processes as mathematical thinking. However, there is only weak and indirect evidence in the current study that geometric reasoning, as its tested here, draws on the same circuits as math.

---

## [Referee Report · Reviewer #3 (Public review)]

Summary:

The authors report converging evidence from behavioral studies as well as several brain-imaging techniques that geometric figures, notably quadrilaterals, are processed differently in visual (lower activation) and spatial (greater) areas of the human brain than representative figures. Comparison of mathematical models to fit activity for geometric figures shows the best fit for abstract geometric features like parallelism and symmetry. The brain areas active for geometric figures are also active in processing mathematical concepts even in blind mathematicians, linking geometric shapes to abstract math concepts. The effects are stronger in adults than in 6-year-old Western children. Similar phenomena do not appear in great apes, suggesting that this is uniquely human and developmental.

Strengths:

Multiple converging techniques of brain imaging and testing of mathematical models showing special status of perception of abstract forms. Careful reasoning at every step of research and presentation of research, anticipating and addressing possible reservations. Connecting these findings to other findings, brain, behavior, and historical/anthropological to suggest broad and important fundamental connections between abstract visual-spatial forms and mathematical reasoning.

Weaknesses:

I have reservations of the authors' use of "symbolic." They seem to interpret "symbolic" as relying on "discrete, exact, rule-based features." Words are generally considered to symbolic (that is their major function), yet words do not meet those criteria. Depictions of objects can be regarded as symbolic because they represent real objects, they are not the same as the object (as Magritte observed). If so then perhaps depictions of quadrilaterals are also symbolic but then they do not differ from depictions of objects on that quality. Relatedly, calling abstract or generalized representations of forms a distinct "language of thought" doesn't seem supportable by the current findings. Minimally, a language has elements that are combined more or less according to rules. The authors present evidence for geometric forms as elements but nowhere is there evidence for combining them into meaningful strings.

Further thoughts

Incidentally, there have been many attempts at constructing visual languages from visual elements combined by rules, that is, mapping meaning to depictions. Many written languages like Egyptian hieroglyphics or Mayan or Chinese, began that way; there are current attempts using emoji. Apparently, mapping sound to discrete letters, alphabets, is more efficient and was invented once but spread. That said, for restricted domains like maps, circuit diagrams, networks, chemical interactions, mathematics, and more, visual "languages" work quite well.

The findings are striking and as such invite speculation about their meaning and limitations. The images of real objects seem to be interpreted as representations of 3D objects as they activate the same visual areas as real objects. By contrast, the images of 2D geometric forms are not interpreted as representations of real objects but rather seemingly as 2D abstractions. It would be instructive to investigate stimuli that are on a continuum from representational to geometric, e. g., real objects that have simple geometric forms like table tops or boxes under various projections or balls or buildings that are rectangular or triangular. Objects differ from geometric forms in many ways: 3D rather than 2D, more complicated shapes; internal features as well as outlines. The geometric figures used are flat, 2-D, but much geometry is 3-D (e. g. cubes) with similar abstract features. The feature space of geometry is more than parallelism and symmetry; angles are important for example. Listing and testing features would be fascinating.

Can we say that mathematical thinking began with the regularities of shapes or with counting, or both? External representations of counting go far back into prehistory; tallies are frequent and wide-spread. Infants are sensitive to number across domains as are other primates (and perhaps other species). Finding overlapping brain areas for geometric forms and number is intriguing but doesn't show how they are related.

Categories are established in part by contrast categories; are quadrilaterals and triangles and circles different categories? As for quadrilaterals, the authors say some are "completely irregular." Not really; they are still quadrilaterals, if atypical. See Eleanor Rosch's insightful work on (visual) categories. One wonders about distinguishing squashed quadrilaterals from squashed triangles.

What in human experience but not the experience of close primates would drive the abstraction of these geometric properties? It's easy to make a case for elaborate brain processes for recognizing and distinguishing things in the world, shared by many species, but the case for brain areas sensitive to abstracting geometric figures is harder. The fact that these areas are active in blind mathematicians and that they are parietal areas suggest that what is important is spatial far more than visual. Could these geometric figures and their abstract properties be connected in some way to behavior, perhaps with fabrication, construction or use of objects? Or with other interactions with complex objects and environments where symmetry and parallelism (and angles and curvature--and weight and size) would be important? Manual dexterity and fabrication also distinguish humans from great apes (quantitatively not qualitatively) and action drives both visual and spatial representations of objects and spaces in the brain. I certainly wouldn't expect the authors to add research to this already packed paper, but raising some of the conceptual issues would contribute to the significance of the paper.

---

## [Author Response]

The following is the authors’ response to the original reviews

**Reviewer #1 (Public review):**
Weakness:I wonder how task difficulty and linguistic labels interact with the current findings. Based on the behavioral data, shapes with more geometric regularities are easier to detect when surrounded by other shapes. Do shape labels that are readily available (e.g., "square") help in making accurate and speedy decisions? Can the sensitivity to geometric regularity in intraparietal and inferior temporal regions be attributed to differences in task difficulty? Similarly, are the MEG oddball detection effects that are modulated by geometric regularity also affected by task difficulty?

We see two aspects to the reviewer’s remarks.

(1) Names for shapes.

On the one hand, is the question of the impact of whether certain shapes have names and others do not in our task. The work presented here is not designed to specifically test the effect of formal western education; however, in previous work (Sablé-Meyer et al., 2021), we noted that the geometric regularity effect remains present even for shapes that do not have specific names, and even in participants who do not have names for them. Thus, we replicated our main effects with both preschoolers and adults that did not attend formal western education and found that our geometric feature model remained predictive of their behavior; we refer the reader to this previous paper for an extensive discussion of the possible role of linguistic labels, and the impact of the statistics of the environment on task performance.

What is more, in our behavior experiments we can discard data from any shape that is has a name in English and run our model comparison again. Doing so diminished the effect size of the geometric feature model, but it remained predictive of human behavior: indeed, if we removed all shapes but kite, rightKite, rustedHinge, hinge and random (i.e., more than half of our data, and shapes for which we came up with names but there are no established names), we nevertheless find that both models significantly correlate with human behavior—see plot in Author response image 1, equivalent of our Fig. 1E with the remaining shapes.

**Author response image 1. sa4fig1:** 

An identical analysis on the MEG leads to two noisy but significant clusters (CNN: 64.0ms to 172.0ms; then 192.0ms to 296.0ms; both p<.001: Geometric Features: 312.0ms to 364.0ms with p=.008). We have improved our manuscript thanks to the reviewer’s observation by adding a figure with the new behavior analysis to the supplementary figures and in the result section of the behavior task. We now refer to these analysis where appropriate:

(intro) “The effect appeared as a human universal, present in preschoolers, first-graders, and adults without access to formal western math education (the Himba from Namibia), and thus seemingly independent of education and of the existence of linguistic labels for regular shapes.”

(behavior results) “Finally, to separate the effect of name availability and geometric features on behavior, we replicated our analysis after removing the square, rectangle, trapezoids, rhombus and parallelogram from our data (Fig. S5D). This left us with five shapes, and an RDM with 10 entries, When regressing it in a GLM with our two models, we find that both models are still significant predictors (p<.001). The effect size of the geometric feature model is greatly reduced, yet remained significantly higher than that of the neural network model (p<.001).”

(meg results) “This analysis yielded similar clusters when performed on a subset of shapes that do not have an obvious name in English, as was the case for the behavior analysis (CNN Encoding: 64.0ms to 172.0ms; then 192.0ms to 296.0ms; both p<.001: Geometric Features: 312.0ms to 364.0ms with p=.008).”

(discussion, end of behavior section) “Previously, we only found such a significant mixture of predictors in uneducated humans (whether French preschoolers or adults from the Himba community, mitigating the possible impact of explicit western education, linguistic labels, and statistics of the environment on geometric shape representation) (Sablé-Meyer et al., 2021).”

Perhaps the referee’s point can also be reversed: we provide a normative theory of geometric shape complexity which has the potential to explain why certain shapes have names: instead of seeing shape names as the cause of their simpler mental representation, we suggest that the converse could occur, i.e. the simpler shapes are the ones that are given names.

(2) Task difficulty

On the other hand is the question of whether our effect is driven by task difficulty. First, we would like to point out that this point could apply to the fMRI task, which asks for an explicit detection of deviants, but does not apply to the MEG experiment. In MEG, participants passively looked at sequences of shapes which, for a given block, comprising many instances of a fixed standard shape and rare deviants–even if they notice deviants, they have no task related to them. Yet two independent findings validated the geometric features model: there was a large effect of geometric regularity on the MEG response to deviants, and the MEG dissimilarity matrix between standard shapes correlated with a model based on geometric features, better than with a model based on CNNs. While the response to rare deviants might perhaps be attributed to “difficulty” (assuming that, in spite of the absence of an explicit task, participants try to spot the deviants and find this self-imposed task more difficult in runs with less regular shapes), it seems very hard to explain the representational similarity analysis (RSA) findings based on difficulty. Indeed, what motivated us to use RSA analysis in both fMRI and MEG was to stop relying on the response to deviants, and use solely the data from standard or “reference” shapes, and model their neural response with theory-derived regressors.

We have updated the manuscript in several places to make our view on these points clearer:

(experiment 4) “This design allowed us to study the neural mechanisms of the geometric regularity effect without confounding effects of task, task difficulty, or eye movements.”

(figure 4, legend) “(A) Task structure: participants passively watch a constant stream of geometric shapes, one per second (presentation time 800ms). The stimuli are presented in blocks of 30 identical shapes up to scaling and rotation, with 4 occasional deviant shape. Participants do not have a task to perform beside fixating.”

**Reviewer #2 (Public review):**
Weakness:Given that the primary take away from this study is that geometric shape information is found in the dorsal stream, rather than the ventral stream there is very little there is very little discussion of prior work in this area (for reviews, see Freud et al., 2016; Orban, 2011; Xu, 2018). Indeed, there is extensive evidence of shape processing in the dorsal pathway in human adults (Freud, Culham, et al., 2017; Konen & Kastner, 2008; Romei et al., 2011), children (Freud et al., 2019), patients (Freud, Ganel, et al., 2017), and monkeys (Janssen et al., 2008; Sereno & Maunsell, 1998; Van Dromme et al., 2016), as well as the similarity between models and dorsal shape representations (Ayzenberg & Behrmann, 2022; Han & Sereno, 2022).

We thank the reviewer for this opportunity to clarify our writing. We want to use this opportunity to highlight that our primary finding is not about whether the shapes of objects or animals (in general) are processed in the ventral versus or the dorsal pathway, but rather about the much more restricted domain of geometric shapes such as squares and triangles. We propose that simple geometric shapes afford additional levels of mental representation that rely on their geometric features – on top of the typical visual processing. To the best of our knowledge, this point has not been made in the above papers.

Still, we agree that it is useful to better link our proposal to previous ones. We have updated the discussion section titled “Two Visual Pathways” to include more specific references to the literature that have reported visual object representations in the dorsal pathway. Following another reviewer’s observation, we have also updated our analysis to better demonstrate the overlap in activation evoked by math and by geometry in the IPS, as well as include a novel comparison with independently published results.

Overall, to address this point, we (i) show the overlap between our “geometry” contrast (shape > word+tools+houses) and our “math” contrast (number > words); (ii) we display these ROIs side by side with ROIs found in previous work (Amalric and Dehaene, 2016), and (iii) in each math-related ROIs reported in that article, we test our “geometry” (shape > word+tools+houses) contrast and find almost all of them to be significant in both population; see Fig. S5.

Finally, within the ROIs identified with our geometry localizer, we also performed similarity analyses: for each region we extracted the betas of every voxel for every visual category, and estimated the distance (cross-validated mahalanobis) between different visual categories. In both ventral ROIs, in both populations, numbers were closer to shapes than to the other visual categories including text and Chinese characters (all p<.001). In adults, this result also holds for the right ITG (p=.021) and the left IPS (p=.014) but not the right IPS (p=.17). In children, this result did not hold in the areas.

Naturally, overlap in brain activation does not suffice to conclude that the same computational processes are involved. We have added an explicit caveat about this point. Indeed, throughout the article, we have been careful to frame our results in a way that is appropriate given our evidence, e.g. saying “Those areas are similar to those active during number perception, arithmetic, geometric sequences, and the processing of high-level math concepts” and “The IPS areas activated by geometric shapes overlap with those active during the comprehension of elementary as well as advanced mathematical concepts”. We have rephrased the possibly ambiguous “geometric shapes activated math- and number-related areas, particular the right aIPS.” into “geometric shapes activated areas independently found to be activated by math- and number-related tasks, in particular the right aIPS”.

**Reviewer #3 (Public review):**
Weakness:Perhaps the manuscript could emphasize that the areas recruited by geometric figures but not objects are spatial, with reduced processing in visual areas. It also seems important to say that the images of real objects are interpreted as representations of 3D objects, as they activate the same visual areas as real objects. By contrast, the images of geometric forms are not interpreted as representations of real objects but rather perhaps as 2D abstractions.

This is an interesting possibility. Geometric shapes are likely to draw attention to spatial dimensions (e.g. length) and to do so in a 2D spatial frame of reference rather than the 3D representations evoked by most other objects or images. However, this possibility would require further work to be thoroughly evaluated, for instance by comparing usual 3D objects with rare instances of 2D ones (e.g. a sheet of paper, a sticker etc). In the absence of such a test, we refrained from further speculation on this point.

The authors use the term "symbolic." That use of that term could usefully be expanded here.

The reviewer is right in pointing out that “symbolic” should have been more clearly defined. We now added in the introduction:

(introduction) “[…] we sometimes refer to this model as “symbolic” because it relies on discrete, exact, rule-based features rather than continuous representations (Sablé-Meyer et al., 2022). In this representational format, geometric shapes are postulated to be represented by symbolic expressions in a “language-of-thought”, e.g. “a square is a four-sided figure with four equal sides and four right angles” or equivalently by a computer-like program from drawing them in a Logo-like language (Sablé-Meyer et al., 2022).”

Here, however, the present experiments do not directly probe this format of a representation. We have therefore simplified our wording and removed many of our use of the word “symbolic” in favor of the more specific “geometric features”.

Pigeons have remarkable visual systems. According to my fallible memory, Herrnstein investigated visual categories in pigeons. They can recognize individual people from fragments of photos, among other feats. I believe pigeons failed at geometric figures and also at cartoon drawings of things they could recognize in photos. This suggests they did not interpret line drawings of objects as representations of objects.

The comparison of geometric abilities across species is an interesting line of research. In the discussion, we briefly mention several lines of research that indicate that non-human primates do not perceive geometric shapes in the same way as we do – but for space reasons, we are reluctant to expand this section to a broader review of other more distant species. The referee is right that there is evidence of pigeons being able to perceive an invariant abstract 3D geometric shape in spite of much variation in viewpoint (Peissig et al., 2019) – but there does not seem to be evidence that they attend to geometric regularities specifically (e.g. squares versus non-squares). Also, the referee’s point bears on the somewhat different issue of whether humans and other animals may recognize the object depicted by a symbolic drawing (e.g. a sketch of a tree). Again, humans seem to be vastly superior in this domain, and research on this topic is currently ongoing in the lab. However, the point that we are making in the present work is specifically about the neural correlates of the representation of simple geometric shapes which by design were not intended to be interpretable as representations of objects.

Categories are established in part by contrast categories; are quadrilaterals, triangles, and circles different categories?

We are not sure how to interpret the referee’s question, since it bears on the definition of “category” (Spontaneous? After training? With what criterion?). While we are not aware of data that can unambiguously answer the reviewer’s question, categorical perception in geometric shapes can be inferred from early work investigating pop-out effects in visual search, e.g. (Treisman and Gormican, 1988): curvature appears to generate strong pop-out effects, and therefore we would expect e.g. circles to indeed be a different category than, say, triangles. Similarly, right angles, as well as parallel lines, have been found to be perceived categorically (Dillon et al., 2019).

This suggests that indeed squares would be perceived as categorically different from triangles and circles. On the other hand, in our own previous work (Sablé-Meyer et al., 2021) we have found that the deviants that we generated from our quadrilaterals did not pop out from displays of reference quadrilaterals. Pop-out is probably not the proper criterion for defining what a “category” is, but this is the extent to which we can provide an answer to the reviewer’s question.

It would be instructive to investigate stimuli that are on a continuum from representational to geometric, e.g., table tops or cartons under various projections, or balls or buildings that are rectangular or triangular. Building parts, inside and out. like corners. Objects differ from geometric forms in many ways: 3D rather than 2D, more complicated shapes, and internal texture. The geometric figures used are flat, 2-D, but much geometry is 3-D (e. g. cubes) with similar abstract features.

We agree that there is a whole line of potential research here. We decided to start by focusing on the simplest set of geometric shapes that would give us enough variation in geometric regularity while being easy to match on other visual features. We agree with the reviewer that our results should hold both for more complex 2-D shapes, but also for 3-D shapes. Indeed, generative theories of shapes in higher dimensions following similar principles as ours have been devised (I. Biederman, 1987; Leyton, 2003). We now mention this in the discussion:

“Finally, this research should ultimately be extended to the representation of 3-dimensional geometric shapes, for which similar symbolic generative models have indeed been proposed (Irving Biederman, 1987; Leyton, 2003).”

The feature space of geometry is more than parallelism and symmetry; angles are important, for example. Listing and testing features would be fascinating. Similarly, looking at younger or preferably non-Western children, as Western children are exposed to shapes in play at early ages.

We agree with the reviewer on all point. While we do not list and test the different properties separately in this work, we would like to highlight that angles are part of our geometric feature model, which includes features of “right-angle” and “equal-angles” as suggested by the reviewer.

We also agree about the importance of testing populations with limited exposure to formal training with geometric shapes. This was in fact a core aspect of a previous article of ours which tests both preschoolers, and adults with no access to formal western education – though no non-Western children (Sablé-Meyer et al., 2021). It remains a challenge to perform brain-imaging studies in non-Western populations (although see Dehaene et al., 2010; Pegado et al., 2014).

What in human experience but not the experience of close primates would drive the abstraction of these geometric properties? It's easy to make a case for elaborate brain processes for recognizing and distinguishing things in the world, shared by many species, but the case for brain areas sensitive to processing geometric figures is harder. The fact that these areas are active in blind mathematicians and that they are parietal areas suggests that what is important is spatial far more than visual. Could these geometric figures and their abstract properties be connected in some way to behavior, perhaps with fabrication and construction as well as use? Or with other interactions with complex objects and environments where symmetry and parallelism (and angles and curvature--and weight and size) would be important? Manual dexterity and fabrication also distinguish humans from great apes (quantitatively, not qualitatively), and action drives both visual and spatial representations of objects and spaces in the brain. I certainly wouldn't expect the authors to add research to this already packed paper, but raising some of the conceptual issues would contribute to the significance of the paper.

We refrained from speculating about this point in the previous version of the article, but share some of the reviewers’ intuitions about the underlying drive for geometric abstraction. As described in (Dehaene, 2026; Sablé-Meyer et al., 2022), our hypothesis, which isn’t tested in the present article, is that the emergence of a pervasive ability to represent aspects of the world as compact expressions in a mental “language-of-thought” is what underlies many domains of specific human competence, including some listed by the reviewer (tool construction, scene understanding) and our domain of study here, geometric shapes.

**Recommendations for the Authors:**

**Reviewer #1 (Recommendations for the authors):**
Overall, I enjoyed reading this paper. It is clearly written and nicely showcases the amount of work that has gone into conducting all these experiments and analyzing the data in sophisticated ways. I also thought the figures were great, and I liked the level of organization in the GitHub repository and am looking forward to seeing the shared data on OpenNeuro. I have some specific questions I hope the authors can address.(1) Behavior- Looking at Figure 1, it seemed like most shapes are clustering together, whereas square, rectangle, and maybe rhombus and parallelogram are slightly more unique. I was wondering whether the authors could comment on the potential influence of linguistic labels. Is it possible that it is easier to discard the intruder when the shapes are readily nameable versus not?

This is an interesting observation, but the existence of names for shapes does not suffice to explain all of our findings ; see our reply to the public comment.

(2) fMRI- As mentioned in the public review, I was surprised that the authors went with an intruder task because I would imagine that performance depends on the specific combination of geometric shapes used within a trial. I assume it is much harder to find, for example, a "Right Hinge" embedded within "Hinge" stimuli than a "Right Hinge" amongst "Squares". In addition, the rotation and scaling of each individual item should affect regular shapes less than irregular shapes, creating visual dissimilarities that would presumably make the task harder. Can the authors comment on how we can be sure that the differences we pick up in the parietal areas are not related to task difficulty but are truly related to geometric shape regularities?

Again, please see our public review response for a larger discussion of the impact of task difficulty. There are two aspects to answering this question.

First, the task is not as the reviewer describes: the intruder task is to find a deviant shape within several slightly rotated and scaled versions of the regular shape it came from. During brain imaging, we did not ask participants to find an exemplar of one of our reference shape amidst copies of another, but rather a deviant version of one shape against copies of its reference version. We only used this intruder task with all pairs of shapes to generate the behavioral RSA matrix.

Second, we agree that some of the fMRI effect may stem from task difficulty, and this motivated our use of RSA analysis in fMRI, and a passive MEG task. RSA results cannot be explained by task difficulty.

Overall, we have tried to make the limitations of the fMRI design, and the motivation for turning to passive presentation in MEG, clearer by stating the issues more clearly when we introduce experiment 4:

“The temporal resolution of fMRI does not allow to track the dynamic of mental representations over time. Furthermore, the previous fMRI experiment suffered from several limitations. First, we studied six quadrilaterals only, compared to 11 in our previous behavioral work. Second, we used an explicit intruder detection, which implies that the geometric regularity effect was correlated with task difficulty, and we cannot exclude that this factor alone explains some of the activations in figure 3C (although it is much less clear how task difficulty alone would explain the RSA results in figure 3D). Third, the long display duration, which was necessary for good task performance especially in children, afforded the possibility of eye movements, which were not monitored inside the 3T scanner and again could have affected the activations in figure 3C.”

- How far in the periphery were the stimuli presented? Was eye-tracking data collected for the intruder task? Similar to the point above, I would imagine that a harder trial would result in more eye movements to find the intruder, which could drive some of the differences observed here.

A 1-degree bar was added to Figure 3A, which faithfully illustrates how the stimuli were presented in fMRI. Eye-tracking data was not collected during fMRI. Although the participants were explicitly instructed to fixate at the center of the screen and avoid eye movements, we fully agree with the referee that we cannot exclude that eye movements were present, perhaps more so for more difficult displays, and would therefore have contributed to the observed fMRI activations in experiment 3 (figure 3C). We now mention this limitation explicity at the end of experiment 3. However, crucially, this potential problem cannot apply to the MEG data. During the MEG task, the stimuli were presented one by one at the center of screen, without any explicit task, thus avoiding issues of eye movements. We therefore consider the MEG geometrical regularity effect, which comes at a relatively early latency (starting at ~160 ms) and even in a passive task, to provide the strongest evidence of geometric coding, unaffected by potential eye movement artefacts.

- I was wondering whether the authors would consider showing some un-thresholded maps just to see how widespread the activation of the geometric shapes is across all of the cortex.

We share the uncorrected threshold maps in Fig. S3. for both adults and children in the category localizer, copied here as well. For the geometry task, most of the clusters identified are fairly big and survive cluster-corrected permutations; the uncorrected statistical maps look almost fully identical to the one presented in Fig. 3 (p<.001 map).

- I'm missing some discussion on the role of early visual areas that goes beyond the RSA-CNN comparison. I would imagine that early visual areas are not only engaged due to top-down feedback (line 258) but may actually also encode some of the geometric features, such as parallel lines and symmetry. Is it feasible to look at early visual areas and examine what the similarity structure between different shapes looks like?

If early visual areas encoded the geometric features that we propose, then even early sensor-level RSA matrices should show a strong impact of geometric features similarity, which is not what we find (figure 4D). We do, however, appreciate the referee’s request to examine more closely how this similarity structure looks like. We now provide a movie showing the significant correlation between neural activity and our two models (uncorrected participants); indeed, while the early occipital activity (around 110ms) is dominated by a significant correlation with the CNN model, there are also scattered significant sources associated to the symbolic model around these timepoints already.

To test this further, we used beamformers to reconstruct the source-localized activity in calcarine cortex and performed an RSA analysis across that ROI. We find that indeed the CNN model is strongly significant at t=110ms (t=3.43, df=18, p=.003) while the geometric feature model is not (t=1.04, df=18, p=.31), and the CNN is significantly above the geometric feature model (t=4.25, df=18, p<.001). However, this result is not very stable across time, and there are significant temporal clusters around these timepoints associated to each model, with no significant cluster associated to a CNN > geometric (CNN: significant cluster from 88ms to 140ms, p<.001 in permutation based with 10000 permutations; geometric features has a significant cluster from 80ms to 104ms, p=.0475; no significant cluster on the difference between the two).

(3) MEG- Similar to the fMRI set, I am a little worried that task difficulty has an effect on the decoding results, as the oddball should pop out more in more geometric shapes, making it easier to detect and easier to decode. Can the authors comment on whether it would matter for the conclusions whether they are decoding varying task difficulty or differences in geometric regularity, or whether they think this can be considered similarly?

See above for an extensive discussion of the task difficulty effect. We point out that there is no task in the MEG data collection part. We have clarified the task design by updating our Fig. 4. Additionally, the fact that oddballs are more perceived more or less easily as a function of their geometric regularity is, in part, exactly the point that we are making – but, in MEG, even in the absence of a task of looking for them.

- The authors discuss that the inflated baseline/onset decoding/regression estimates may occur because the shapes are being repeated within a mini-block, which I think is unlikely given the long ISIs and the fact that the geometric features model is not >0 at onset. I think their second possible explanation, that this may have to do with smoothing, is very possible. In the text, it said that for the non-smoothed result, the CNN encoding correlates with the data from 60ms, which makes a lot more sense. I would like to encourage the authors to provide readers with the unsmoothed beta values instead of the 100-ms smoothed version in the main plot to preserve the reason they chose to use MEG - for high temporal resolution!

We fully agree with the reviewer and have accordingly updated the figures to show the unsmoothed data (see below). Indeed, there is now no significant CNN effect before ~60 ms (up to the accuracy of identifying onsets with our method).

- In Figure 4C, I think it would be useful to either provide error bars or show variability across participants by plotting each participant's beta values. I think it would also be nice to plot the dissimilarity matrices based on the MEG data at select timepoints, just to see what the similarity structure is like.

Following the reviewer’s recommendation, we plot the timeseries with SEM as shaded area, and thicker lines for statistically significant clusters, and we provide the unsmoothed version in figure Fig. 4. As for the dissimilarity matrices at select timepoints, this has now been added to figure Fig. 4.

- To evaluate the source model reconstruction, I think the reader would need a little more detail on how it was done in the main text. How were the lead fields calculated? Which data was used to estimate the sources? How are the models correlated with the source data?

We have imported some of the details in the main text as follows (as well as expanding the methods section a little):

“To understand which brain areas generated these distinct patterns of activations, and probe whether they fit with our previous fMRI results, we performed a source reconstruction of our data. We projected the sensor activity onto each participant's cortical surfaces estimated from T1-images. The projection was performed using eLORETA and emptyroom recordings acquired on the same day to estimate noise covariance, with the default parameters of mne-bids-pipeline. Sources were spaced using a recursively subdivided octahedron (oct5). Group statistics were performed after alignement to fsaverage. We then replicated the RSA analysis […]”

- In addition to fitting the CNN, which is used here to model differences in early visual cortex, have the authors considered looking at their fMRI results and localizing early visual regions, extracting a similarity matrix, and correlating that with the MEG and/or comparing it with the CNN model?

We had ultimately decided against comparing the empirical similarity matrices from the MEG and fMRI experiments, first because the stimuli and tasks are different, and second because this would not be directly relevant to our goal, which is to evaluate whether a geometric-feature model accounts for the data. Thus, we systematically model empirical similarity matrices from fMRI and from MEG with our two models derived from different theories of shape perception in order to test predictions about their spatial and temporal dynamic. As for comparing the similarity matrix from early visual regions in fMRI with that predicted by the CNN model, this is effectively visible from our Fig. 3D where we perform searchlight RSA analysis and modeling with both the CNN and the geometric feature model; bilaterally, we find a correlation with the CNN model, although it sometimes overlap with predictions from the geometric feature model as well. We now include a section explaining this reasoning in appendix:

“Representational similarity analysis also offers a way to directly compared similarity matrices measured in MEG and fMRI, thus allowing for fusion of those two modalities and tentatively assigning a “time stamp” to distinct MRI clusters. However, we did not attempt such an analysis here for several reasons. First, distinct tasks and block structures were used in MEG and fMRI. Second, a smaller list of shapes was used in fMRI, as imposed by the slower modality of acquisition. Third, our study was designed as an attempt to sort out between two models of geometric shape recognition. We therefore focused all analyses on this goal, which could not have been achieved by direct MEG-fMRI fusion, but required correlation with independently obtained model predictions.”

Minor comments- It's a little unclear from the abstract that there is children's data for fMRI only.

We have reworded the abstract to make this unambiguous

- Figures 4a & b are missing y-labels.

We can see how our labels could be confused with (sub-)plot titles and have moved them to make the interpretation clearer.

- MEG: are the stimuli always shown in the same orientation and size?

They are not, each shape has a random orientation and scaling. On top of a task example at the top of Fig. 4, we have now included a clearer mention of this in the main text when we introduce the task:

“shapes were presented serially, one at a time, with small random changes in rotation and scaling parameters, in miniblocks with a fixed quadrilateral shape and with rare intruders with the bottom right corner shifted by a fixed amount (Sablé-Meyer et al., 2021)”

- To me, the discussion section felt a little lengthy, and I wonder whether it would benefit from being a little more streamlined, focused, and targeted. I found that the structure was a little difficult to follow as it went from describing the result by modality (behavior, fMRI, MEG) back to discussing mostly aspects of the fMRI findings.

We have tried to re-organize and streamline the discussion following these comments.

Then, later on, I found that especially the section on "neurophysiological implementation of geometry" went beyond the focus of the data presented in the paper and was comparatively long and speculative.

We have reexamined the discussion, but the citation of papers emphasizing a representation of non-accidental geometric properties in non-human animals was requested by other commentators on our article; and indeed, we think that they are relevant in the context of our prior suggestion that the composition of geometric features might be a uniquely human feature – these papers suggest that individual features may not, and that it is therefore compositionality which might be special to the human brain. We have nevertheless shortened it.

Furthermore, we think that this section is important because symbolic models are often criticized for lack of a plausible neurophysiological implementation. It is therefore important to discuss whether and how the postulated symbolic geometric code could be realized in neural circuits. We have added this justification to the introduction of this section.

**Reviewer #2 (Recommendations for the authors):**
(1) If the authors want to specifically claim that their findings align with mathematical reasoning, they could at least show the overlap between the activation maps of the current study and those from prior work.

This was added to the fMRI results. See our answers to the public review.

(2) I wonder if the reason the authors only found aIPS in their first analysis (Figure 2) is because they are contrasting geometric shapes with figures that also have geometric properties. In other words, faces, objects, and houses also contain geometric shape information, and so the authors may have essentially contrasted out other areas that are sensitive to these features. One indication that this may be the case is that the geometric regularity effect and searchlight RSA (Figure 3) contains both anterior and posterior IPS regions (but crucially, little ventral activity). It might be interesting to discuss the implications of these differences.

Indeed, we cannot exclude that the few symmetries, perpendicularity and parallelism cues that can be presented in faces, objects or houses were processed as such, perhaps within the ventral pathway, and that these representations would have been subtracted out. We emphasize that our subtraction isolates the geometrical features that are present in simple regular geometric shapes, over and above those that might exist in other categories. We have added this point to the discussion:

“[…] For instance, faces possess a plane of quasi-symmetry, and so do many other man-made tools and houses. Thus, our subtraction isolated the geometrical features that are present in simple regular geometric shapes (e.g. parallels, right angles, equality of length) over and above those that might already exist, in a less pure form, in other categories.”

(3) I had a few questions regarding the MEG results.a. I didn't quite understand the task. What is a regular or oddball shape in this context? It's not clear what is being decoded. Perhaps a small example of the MEG task in Figure 4 would help?

We now include an additional sub-figure in Fig. 4 to explain the paradigm. In brief: there is no explicit task, participants are simply asked to fixate. The shapes come in miniblocks of 30 identical reference shapes (up to rotation and scaling), among which some occasional deviant shapes randomly appear (created by moving the corner of the reference shape by some amount).

b. In Figure 4A/B they describe the correlation with a 'symbolic model'. Is this the same as the geometric model in 4C?

It is. We have removed this ambiguity by calling it “geometric model” and setting its color to the one associated to this model thought the article.

c. The author's explanation for why geometric feature coding was slower than CNN encoding doesn't quite make sense to me. As an explanation, they suggest that previous studies computed "elementary features of location or motor affordance", whereas their study work examines "high-level mathematical information of an abstract nature." However, looking at the studies the authors cite in this section, it seems that these studies also examined the time course of shape processing in the dorsal pathway, not "elementary features of location or motor affordance." Second, it's not clear how the geometric feature model reflects high-level mathematical information (see point above about claiming this is related to math).

We thank the referee for pointing out this inappropriate phrase, which we removed. We rephrased the rest of the paragraph to clarify our hypothesis in the following way:

“However, in this work, we specifically probed the processing of geometric shapes that, if our hypothesis is correct, are represented as mental expressions that combine geometrical and arithmetic features of an abstract categorical nature, for instance representing “four equal sides” or “four right angles”. It seems logical that such expressions, combining number, angle and length information, take more time to be computed than the first wave of feedforward processing within the occipito-temporal visual pathway, and therefore only activate thereafter.”

One explanation may be that the authors' geometric shapes require finer-grained discrimination than the object categories used in prior studies. i.e., the odd-ball task may be more of a fine-grained visual discrimination task. Indeed, it may not be a surprise that one can decode the difference between, say, a hammer and a butterfly faster than two kinds of quadrilaterals.

We do not disagree with this intuition, although note that we do not have data on this point (we are reporting and modelling the MEG RSA matrix across geometric shapes only – in this part, no other shapes such as tools or faces are involved). Still, the difference between squares, rectangles, parallelograms and other geometric shapes in our stimuli is not so subtle. Furthermore, CNNs do make very fine grained distinctions, for instance between many different breeds of dogs in the IMAGENET corpus. Still, those sorts of distinctions capture the initial part of the MEG response, while the geometric model is needed only for the later part. Thus, we think that it is a genuine finding that geometric computations associated with the dorsal parietal pathway are slower than the image analysis performed by the ventral occipito-temporal pathway.

d. CNN encoding at time 0 is a little weird, but the author's explanation, that this is explained by the fact that temporal smoothed using a 100 ms window makes sense. However, smoothing by 100 ms is quite a lot, and it doesn't seem accurate to present continuous time course data when the decoding or RSA result at each time point reflects a 100 ms bin. It may be more accurate to simply show unsmoothed data. I'm less convinced by the explanation about shape prediction.

We agree. Following the reviewer’s advice, as well as the recommendation from reviewer 1, we now display unsmoothed plots, and the effects now exhibit a more reasonable timing (Figure 4D), with effects starting around ~60 ms for CNN encoding.

(4) I appreciate the author's use of multiple models and their explanation for why DINOv2 explains more variance than the geometric and CNN models (that it represents both types of features. A variance partitioning analysis may help strengthen this conclusion Bonner & Epstein, 2018; Lescroart et al., 2015).However, one difference between DINOv2 and the CNN used here is that it is trained on a dataset of 142 million images vs. the 1.5 million images used in ImageNet. Thus, DINOv2 is more likely to have been exposed to simple geometric shapes during training, whereas standard ImageNet trained models are not. Indeed, prior work has shown that lesioning line drawing-like images from such datasets drastically impairs the performance of large models (Mayilvahanan et al., 2024). Thus, it is unlikely that the use of a transformer architecture explains the performance of DINOv2. The authors could include an ImageNet-trained transformer (e.g., ViT) and a CNN trained on large datasets (e.g., ResNet trained on the Open Clip dataset) to test these possibilities. However, I think it's also sufficient to discuss visual experience as a possible explanation for the CNN and DINOv2 results. Indeed, young children are exposed to geometric shapes, whereas ImageNet-trained CNNs are not.

We agree with the reviewer’s observation. In fact, new and ongoing work from the lab is also exploring this; we have included in supplementary materials exactly what the reviewer is suggesting, namely the time course of the correlation with ViT and with ConvNeXT. In line with the reviewers’ prediction, these networks, trained on much larger dataset and with many more parameters, can also fit the human data as well as DINOv2. We ran additional analysis of the MEG data with ViT and ConvNeXT, which we now report in Fig. S6 as well as in an additional sentence in that section:

“[…] similar results were obtained by performing the same analysis, not only with another vision transformer network, ViT, but crucially using a much larger convolutional neural network, ConvNeXT, which comprises ~800M parameters and has been trained on 2B images, likely including many geometric shapes and human drawings. For the sake of completeness, RSA analysis in sensor space of the MEG data with these two models is provided in Fig. S6.”

We conclude that the size and nature of the training set could be as important as the architecture – but also note that humans do not rely on such a huge training set. We have updated the text, as well as Fig. S6, accordingly by updating the section now entitled “Vision Transformers and Larger Neural Networks”, and the discussion section on theoretical models.

(5) The authors may be interested in a recent paper from Arcaro and colleagues that showed that the parietal cortex is greatly expanded in humans (including infants) compared to non-human primates (Meyer et al., 2025), which may explain the stronger geometric reasoning abilities of humans.

A very interesting article indeed! We have updated our article to incorporate this reference in the discussion, in the section on visual pathways, as follows:

“Finally, recent work shows that within the visual cortex, the strongest relative difference in growth between human and non-human primates is localized in parietal areas (Meyer et al., 2025). If this expansion reflected the acquisition of new processing abilities in these regions, it might explain the observed differences in geometric abilities between human and non-human primates (Sablé-Meyer et al., 2021).”

Also, the authors may want to include this paper, which uses a similar oddity task and compelling shows that crows are sensitive to geometric regularity:Schmidbauer, P., Hahn, M., & Nieder, A. (2025). Crows recognize geometric regularity. Science Advances, 11(15), eadt3718. https://doi.org/10.1126/sciadv.adt3718

We have ongoing discussions with the authors of this work and are have prepared a response to their findings (Sablé-Meyer and Dehaene, 2025)–ultimately, we think that this discussion, which we agree is important, does not have its place in the present article. They used a reduced version of our design, with amplified differences in the intruders. While they did not test the fit of their model with CNN or geometric feature models, we did and found that a simple CNN suffices to account for crow behavior. Thus, we disagree that their conclusions follow from their results and their conclusions. But the present article does not seem to be the right platform to engage in this discussion.

References

Ayzenberg, V., & Behrmann, M. (2022). The Dorsal Visual Pathway Represents Object-Centered Spatial Relations for Object Recognition. The Journal of Neuroscience, 42(23), 4693-4710. https://doi.org/10.1523/jneurosci.2257-21.2022

Bonner, M. F., & Epstein, R. A. (2018). Computational mechanisms underlying cortical responses to the affordance properties of visual scenes. PLoS Computational Biology, 14(4), e1006111. https://doi.org/10.1371/journal.pcbi.1006111

Bueti, D., & Walsh, V. (2009). The parietal cortex and the representation of time, space, number and other magnitudes. Philosophical Transactions of the Royal Society B: Biological Sciences, 364(1525), 1831-1840.

Dehaene, S., & Brannon, E. (2011). Space, time and number in the brain: Searching for the foundations of mathematical thought. Academic Press.

Freud, E., Culham, J. C., Plaut, D. C., & Bermann, M. (2017). The large-scale organization of shape processing in the ventral and dorsal pathways. eLife, 6, e27576.

Freud, E., Ganel, T., Shelef, I., Hammer, M. D., Avidan, G., & Behrmann, M. (2017). Three-dimensional representations of objects in dorsal cortex are dissociable from those in ventral cortex. Cerebral Cortex, 27(1), 422-434.

Freud, E., Plaut, D. C., & Behrmann, M. (2016). 'What 'is happening in the dorsal visual pathway. Trends in Cognitive Sciences, 20(10), 773-784.

Freud, E., Plaut, D. C., & Behrmann, M. (2019). Protracted developmental trajectory of shape processing along the two visual pathways. Journal of Cognitive Neuroscience, 31(10), 1589-1597.

Han, Z., & Sereno, A. (2022). Modeling the Ventral and Dorsal Cortical Visual Pathways Using Artificial Neural Networks. Neural Computation, 34(1), 138-171. https://doi.org/10.1162/neco_a_01456

Janssen, P., Srivastava, S., Ombelet, S., & Orban, G. A. (2008). Coding of shape and position in macaque lateral intraparietal area. Journal of Neuroscience, 28(26), 6679-6690.

Konen, C. S., & Kastner, S. (2008). Two hierarchically organized neural systems for object information in human visual cortex. Nature Neuroscience, 11(2), 224-231.

Lescroart, M. D., Stansbury, D. E., & Gallant, J. L. (2015). Fourier power, subjective distance, and object categories all provide plausible models of BOLD responses in scene-selective visual areas. Frontiers in Computational Neuroscience, 9(135), 1-20. https://doi.org/10.3389/fncom.2015.00135

Mayilvahanan, P., Zimmermann, R. S., Wiedemer, T., Rusak, E., Juhos, A., Bethge, M., & Brendel, W. (2024). In search of forgotten domain generalization. arXiv Preprint arXiv:2410.08258.

Meyer, E. E., Martynek, M., Kastner, S., Livingstone, M. S., & Arcaro, M. J. (2025). Expansion of a conserved architecture drives the evolution of the primate visual cortex. Proceedings of the National Academy of Sciences, 122(3), e2421585122. https://doi.org/10.1073/pnas.2421585122

Orban, G. A. (2011). The extraction of 3D shape in the visual system of human and nonhuman primates. Annual Review of Neuroscience, 34, 361-388.

Romei, V., Driver, J., Schyns, P. G., & Thut, G. (2011). Rhythmic TMS over Parietal Cortex Links Distinct Brain Frequencies to Global versus Local Visual Processing. Current Biology, 21(4), 334-337. https://doi.org/10.1016/j.cub.2011.01.035

Sereno, A. B., & Maunsell, J. H. R. (1998). Shape selectivity in primate lateral intraparietal cortex. Nature, 395(6701), 500-503. https://doi.org/10.1038/26752

Summerfield, C., Luyckx, F., & Sheahan, H. (2020). Structure learning and the posterior parietal cortex. Progress in Neurobiology, 184, 101717. https://doi.org/10.1016/j.pneurobio.2019.101717

Van Dromme, I. C., Premereur, E., Verhoef, B.-E., Vanduffel, W., & Janssen, P. (2016). Posterior Parietal Cortex Drives Inferotemporal Activations During Three-Dimensional Object Vision. PLoS Biology, 14(4), e1002445. https://doi.org/10.1371/journal.pbio.1002445

Xu, Y. (2018). A tale of two visual systems: Invariant and adaptive visual information representations in the primate brain. Annu. Rev. Vis. Sci, 4, 311-336.

**Reviewer #3 (Recommendations for the authors):**
Bring into the discussion some of the issues outlined above, especially (a) the spatial rather than visual of the geometric figures and (b) the non-representational aspects of geometric form aspects.

We thank the reviewer for their recommendations – see our response to the public review for more details.